# Sensorimotor transformation of number in the primate parietal cortex

**Laura E. Seidler** [ORCID]**, Stephanie Westendorff & Andreas Nieder** [ORCID] ✉

The neuronal mechanisms by which the brain flexibly transforms perceived numerical values into corresponding numbers of self-generated actions remain poorly understood. Here, we investigated this sensorimotor transformation process in the parietal cortex of two male rhesus macaques performing a manual counting task. Monkeys viewed visual numerical cues and produced a corresponding number of hand movements. Single-neuron recordings from the ventral intraparietal area (VIP)—a region known to represent perceived numerosity—revealed tuning to the number of intended actions during motor planning. These neurons showed both sustained and transient activity patterns, reflecting static and dynamic codes that support numerical sensorimotor transformation. Population decoding confirmed that VIP encoded intended action number and reflected systematic over- and underestimation errors. Our findings reveal a neural mechanism by which the primate brain converts abstract numerical input into goal-directed motor output, providing insight into the sensorimotor foundations of numerical cognition.

Humans and nonhuman primates share an evolutionarily conserved, non-symbolic number sense that supports the intuitive estimation of the number of items, or numerosity[1,2]. This ability is underpinned by a dedicated cortical network centered in the posterior parietal cortex, particularly the intraparietal sulcus (IPS), where numerosity-selective responses have been observed across species[3–5]. Within this network, the ventral intraparietal area (VIP) located in the fundus of the IPS has emerged as a key region encoding abstract numerical values across sensory modalities. VIP neurons selectively tune to specific numerosities, mirroring behaviorally relevant numerical representations[6–9].

While much of the existing research has focused on the perceptual encoding of number, numerical information is also actively employed in goal-directed behavior. For instance, recognizing that 4 days have passed, and then drawing four tally marks demonstrates the transformation of perceived quantity into a sequence of motor actions[10,11]. Although neurons in the superior parietal lobule have been shown to encode the number of movements in monkeys trained to perform fixed five-action sequences, the absence of sensory cues for varying target numerosities in that study limits its relevance to perceptually guided actions[12]. Thus, how the brain flexibly maps varying numerical inputs onto corresponding motor outputs remains unknown.

Psychophysical adaptation studies point to a shared sensorimotor number system—likely situated in the parietal cortex—that integrates numerical perception with action planning[13–16]. Here, we test the hypothesis that neurons in the VIP encode the transformation of perceived numerosity into the number of planned motor actions, providing a neural substrate for sensorimotor integration of numerical information.

## Results

We trained two male rhesus monkeys (*Macaca mulatta*) to perform a manual counting task, in which they evaluated variable visual numerical instruction cues and flexibly translated these percepts into a matching number of self-generated hand movements (Fig. 1A). The instruction stimuli conveyed numerical values from one to five, presented either as dot arrays or as Arabic numerals, with the latter format having been learned by the animals during training to represent specific quantities. For both dot and sign formats, we implemented standard and control stimulus conditions to avoid reliance on non-numerical visual cues in the dot format and to encourage generalization across varying visual appearances within each format (Fig. 1B).

Following the instruction stimulus, a delay period served as a motor planning phase during which the monkeys translated the

Animal Physiology Unit, Institute of Neurobiology, University of Tübingen, Tübingen, Germany. ✉e-mail: andreas.nieder@uni-tuebingen.de

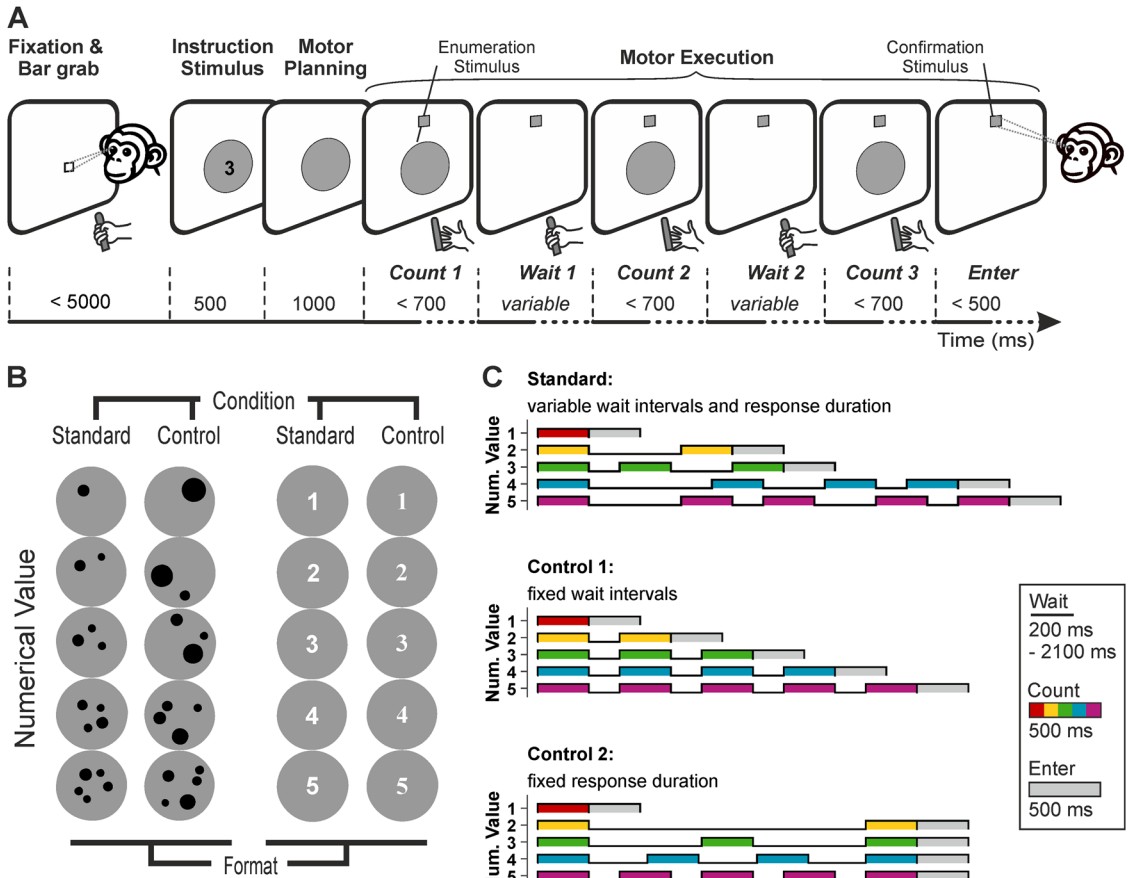

**Fig. 1 | Number production task. A** Outline of the number production task. The trial began when the monkey initiated fixation and held onto a handle. Following the presentation of the instruction stimulus (500 ms), which cued a variable numerical value from 1 to 5, and a motor planning period (1000 ms), the monkey was required to produce the instructed number by repeatedly releasing and holding the handle in response to predetermined, variable-timed enumeration stimuli. Once the monkey judged it had reached the required number of hand movements during the motor execution phase, it shifted its gaze to the confirmation stimulus to signal the end of the counting sequence. **B** Example instruction stimuli. The numerical values 1–5 were instructed using two formats:

dot arrays and signs (Arabic numerals). Within each format, there were control and standard conditions designed to account for non-numerical factors—such as position, size, density, and area in the dot arrays—and for shape appearance (font types) in the sign format. **C** Temporal arrangement of the response period. The monkey's responses in the motor execution period were timed according to three predefined temporal sequence arrangements (every colored bar represents one 'count' event; see **A**): standard, control 1, and control 2, to prevent responses based on timing rather than enumeration. In each session, the standard arrangement was combined with one of the control arrangements (alternating daily) in a pseudo-randomized trial order.

perceived numerical value into a planned sequence of corresponding hand movements. The monkeys responded by releasing a handle the number of times corresponding to the instructed value, with each release permitted only upon the timed appearance of an enumeration stimulus. To prevent reliance on timing-based strategies, the interval between enumeration stimuli varied across three predefined temporal sequence patterns (Fig. 1C). After completing each numeric sequence of hand movements, the monkeys indicated the end of the count by shifting their gaze from the central fixation point to a confirmation stimulus—present throughout the response phase—serving as an enter key. Note that the temporal arrangement of possible hand movements on any given trial was unknown to the monkeys and therefore could not have influenced neuronal activity during the motor planning period. The trial was only completed upon the gaze shift to the confirmation stimulus, and the monkeys were not limited in the number of hand movements they could make.

**Behavioral performance**

Both monkeys accurately performed the number production task, releasing the handle the instructed number of times as indicated by the cue (Monkey 1: 60.6 ± 5.0% correct, $n = 36$ sessions; average 415 correct trials per session; Monkey 2: 56.3 ± 7.5%, $n = 40$, 221 correct

trials per session) (Fig. 2A, B). Performance across sessions for each numerical value, stimulus format, stimulus condition, and temporal arrangement was significantly above chance in both monkeys (one-sided $t$-tests against a chance level of 12.5%, chance level adjusted based on the actual number of actions executed (see "Methods" for details); all $p < 0.001$). Both monkeys performed better with sign formats (Table 1). Monkey 2 showed higher percent-correct values across all five numbers, whereas Monkey 1 exhibited the same pattern except for number 1.

The behavioral response functions depict the frequency of hand movements performed by the monkeys for each instructed number (Fig. 2C, D). For each numerical value across both stimulus formats and temporal conditions, response frequencies peaked at the instructed numerosity. Most errors—either too many or too few releases—occurred near the target, predominantly at adjacent numerical values. Error frequency decreased as the distance from the target number increased, illustrating the numerical distance effect. Additionally, response functions broadened with increasing numerosity, reflecting reduced numerical discriminability known as the numerical size effect. These two effects are well-established characteristics of the approximate number system (ANS) in sensory numerosity judgment[2].

## Single VIP neurons show selectivity for number in the planning period

We recorded the activity of 247 single neurons in VIP while the monkeys performed the number production task (Fig. 3A). During the instruction stimulus phase, and even more during the motor planning phase, many neurons exhibited systematic activity changes in response to numerical values, irrespective of the formats. To identify numerosity-selective neurons and their time intervals with significant modulation driven by number, we applied a two-factor sliding window Analysis of Variance (ANOVA) with the main factors number and

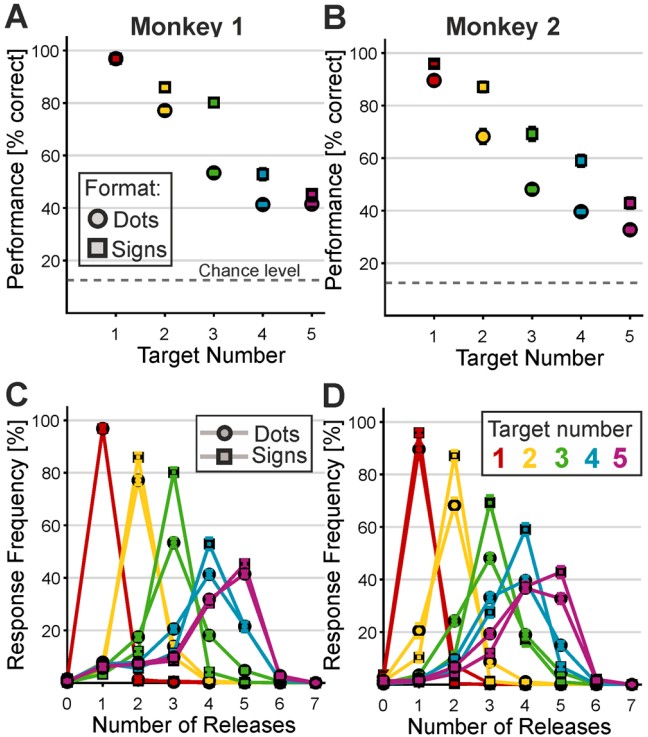

**Fig. 2 | Behavioral performance.** Average performance in % correct responses of monkey 1 (**A**) and monkey 2 (**B**). Performance is depicted separately for the numerical values and formats: The chance level is indicated by the grey dotted line. Averaged over $n = 36$ sessions for monkey 1 and $n = 40$ sessions for monkey 2. Error bars represent the SEM across sessions. Response functions of monkey 1 (**C**) and monkey 2 (**D**) for each instructed numerical value (each color refers to one number) split up for dot (circle) and sign (square) formats. The highest response probability corresponds to the instructed stimulus numerosity. Averaged over $n = 36$ sessions for monkey 1 and $n = 40$ sessions for monkey 2. Error bars represent the SEM across sessions (Source data are provided as a Source Data file).

format (significance threshold $\alpha = 0.01$) (Fig. 3B). Table 2 lists the numbers of neurons selective to main factors and interactions in the instruction stimulus and motor preparation phases. The proportion of selective neurons during the sensory instruction phase (approximately 10%) corresponds to previous findings[17–21]. In the instruction stimulus phase, 12 numerosity-selective neurons were classified as excited and 9 as inhibited. In the motor preparation phase, 56 neurons were excited and 29 were inhibited.

We first focused on the motor planning period during which monkeys translated the perceived numerical value into a corresponding number of self-generated hand movements (Fig. 1A). The temporal arrangement for the subsequent motor execution period on each trial (Fig. 1C) was unknown to the monkeys and could not have influenced neuronal activity during motor planning. We found that over one-third of VIP neurons (34.4%; 85/247) were exclusively number-selective, showing a significant main effect of number without selectivity for format or interactions (Monkey 1: 32.2%; 59/183 neurons; Monkey 2: 40.6%; 26/64 neurons). The onset and duration of selective response periods were highly variable and independent of each neuron's preferred numerical value and were distributed across, collectively spanning, the entire motor planning phase (Fig. 3B). Splitting the data into dot and sign trials confirms that the response windows are reliable (Supplementary Fig. S1). The onset of selectivity for dots versus signs was indifferent in both the instruction phase (median dots: 250 ms; signs: 300 ms; Wilcoxon sign ranks test, $p = 0.94$) and the motor preparation phase (dots: 130 ms; signs: 170 ms; Wilcoxon sign ranks test, $p = 1$). The numerical value eliciting the highest firing rate within the selected time window was defined as each neuron's preferred number. Firing rates decreased systematically with increasing numerical distance from this preferred value. Individually tuned neurons were identified for each instructed number (Fig. 3C–G). Collectively, the tuned neurons spanned the full number range from 1 to 5 (Fig. 4B). Population tuning responses were generated by normalizing and averaging each neuron's tuning curve relative to its preferred number (Fig. 4A). The resulting activity showed overlapping neuronal tuning across the full range of target values, mirroring behavioral performance and reflecting both numerical effects observed in behavior (Fig. 2C, D). Split-half cross-validation of tuning curves confirmed reliable determination of tuning functions and preferred numerosities (Supplementary Fig. S2).

### Population of VIP neurons represents the impending number of hand movements

To assess whether the entire recorded neuronal population, regardless of individual tuning, encoded the planned number of actions, we employed support vector machine (SVM) classifiers. SVMs were trained on neuronal activity during the motor planning period and

## Table 1 | Performance of both monkeys to dot numerosities and associated signs

| Monkey 1 | | | | | | |
|---|---|---|---|---|---|---|
| **Format** | **Number 1** | **Number 2** | **Number 3** | **Number 4** | **Number 5** | **AVG** |
| Signs | 96.8 ± 1.6 | 86.0 ± 1.6 | 80.2 ± 1.7 | 52.9 ± 2.2 | 45.3 ± 1.4 | 66.6 ± 1.1 |
| Dots | 96.9 ± 1.3 | 77.1 ± 1.7 | 53.4 ± 1.7 | 41.3 ± 1.5 | 41.5 ± 1.2 | 55.7 ± 0.8 |
| Statistics ($p$) | 0.96 | $5.39 \times 10^{-5}$ | $2.45 \times 10^{-14}$ | $1.42 \times 10^{-7}$ | 0.025 | $1.09 \times 10^{-11}$ |
| **Monkey 2** | | | | | | |
| **Format** | **Number 1** | **Number 2** | **Number 3** | **Number 4** | **Number 5** | **AVG** |
| Signs | 95.9 ± 1.1 | 87.1 ± 1.9 | 69.2 ± 2.6 | 59.0 ± 2.2 | 42.9 ± 1.8 | 65.8 ± 1.6 |
| Dots | 89.6 ± 2 | 68.2 ± 2.8 | 48.1 ± 1.8 | 39.9 ± 1.6 | 32.7 ± 1.8 | 50.0 ± 1.3 |
| Statistics ($p$) | 0.0019 | $3.73 \times 10^{-8}$ | $1.3 \times 10^{-8}$ | $8.29 \times 10^{-9}$ | $3.53 \times 10^{-6}$ | $7.6 \times 10^{-14}$ |

Average percent correct per numerosity (1–5) and dot-vs-sign format. AVG is average percentages. Based on a total of $n = 36$ session averages. Differences tested with two-tailed $t$-test.
Average percent correct per numerosity (1–5) and dot-vs-sign format. AVG is average percentages. Based on a total of $n = 40$ session averages. Differences tested with two-tailed $t$-test.

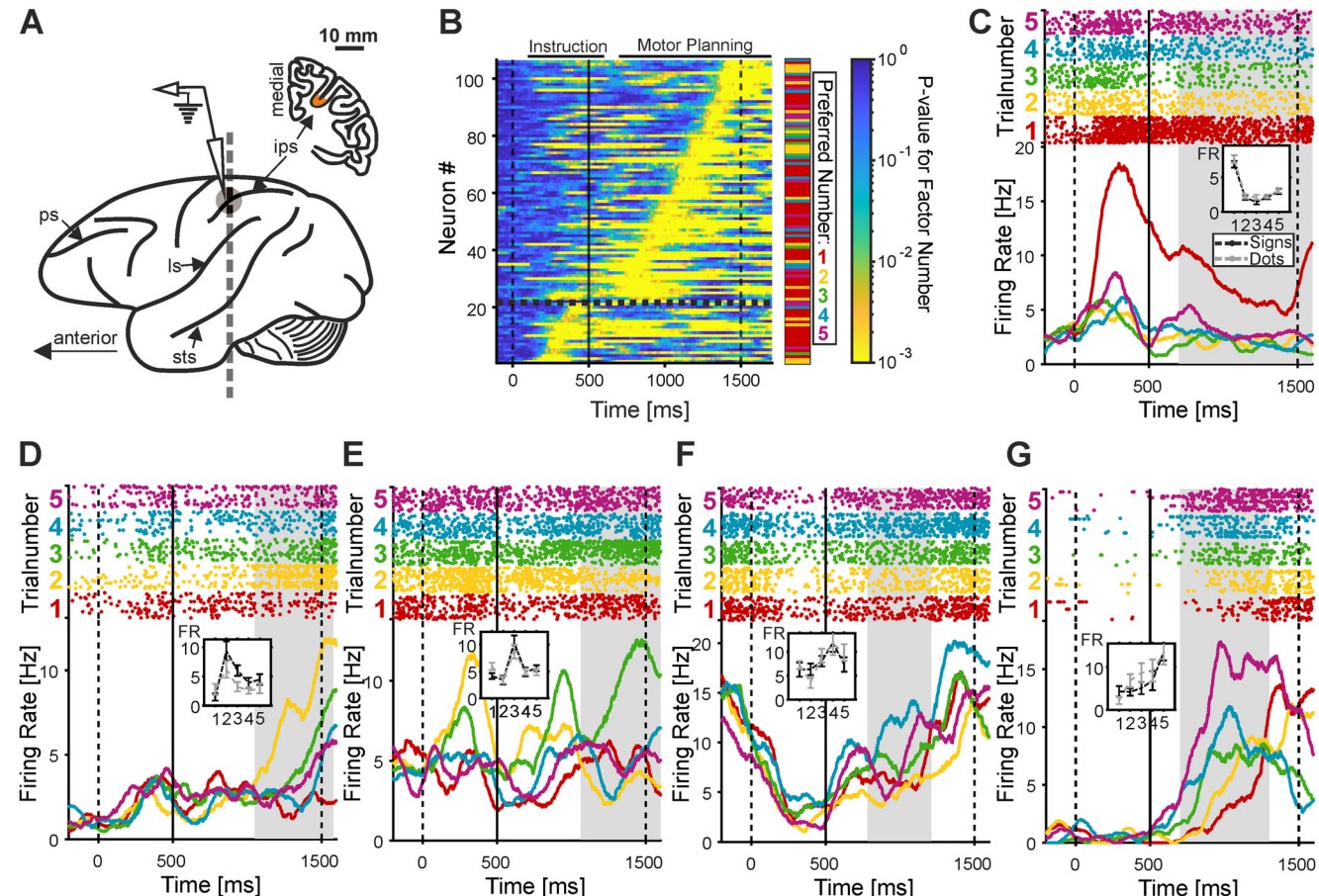

**Fig. 3 | Single VIP neurons show selectivity for number in the planning period.**
**A** Recording site. Lateral view of the left hemisphere of a macaque monkey brain with a coronal section view at the position of the intraparietal sulcus area. Recording site is indicated by grey shading in the fundus of the intraparietal sulcus, area VIP. sts, superior temporal sulcus; ls, lateral sulcus; ips intraparietal sulcus; ps parietal sulcus. **B** Time intervals of single neurons during the instruction stimulus phase (0–500 ms) and motor planning phase (500–1500 ms) when neurons were selective for numerical values in both formats. Each line represents the activity of one neuron (n = 106; Table 2), with surface color indicating the p-value of selectivity. The thick solid lines on top of the surface plot delineates the sliding-window ANOVA analysis interval for the respective phases (accounting for neuronal response latencies). The preferred numerical value of each selective neuron (each

row) is indicated by the color bar on the right corresponding to its target number (1–5). (Source data are provided as a Source Data file). **C–G** Exemplary neurons selective to numerical values 1–5. Dot raster histograms (each dot representing one action potential) are shown in the top panels, with the bottom panels displaying spike-density functions (smoothed with a 150-ms Gaussian kernel). Responses to specific numerical values are color-coded. Insets show the average tuning curves (for the dot and sign formats) of the respective neuron during its selective trial interval (indicated by the shaded area in the histograms, corresponding to the periods of significant selectivity in **B**). Error bars represent the SEM across trials (n = 343, 277, 318, 177, 157 trials for neurons in **C–G**, respectively). Neurons are tuned to one (**C**), two (**D**), three (**E**), four (**F**), and five (**G**) impending numbers of hand movements. FR: firing rate.

**Table 2 | VIP neurons selective according to ANOVA factors**

| Trial phase | Main factor "number" | Main factor "format" | Main factor interaction | Exclusively number selective |
|---|---|---|---|---|
| Instruction stimulus | 9.7% (24/247) | 2.83% (7/247) | 1.2% (3/247) | 8.5% (21/247) |
| Motor preparation | 34.82% (86/247) | 0.81% (2/247) | 1.62% (4/247) | 34.4% (85/247) |

Percentage of selective neurons relative to the total number of n = 247 neurons; two-factor ANOVA, with a significance threshold α = 0.01.

then tested on new trials to classify the planned target number (see "Methods"). First, we tested both stimulus formats combined. The classifier reliably decoded the monkeys' impending target number from population activity, achieving an accuracy of 66.5 ± 0.1% (chance level of 20%). The confusion matrix (Fig. 5A) shows classification accuracy for each target number, with above-chance performance across all values (Number 1: 99.6%, 2: 86.3%, 3: 55.9%, 4: 41.3%, 5: 49.3%; n = 246 neurons). Misclassification errors primarily occurred for numbers adjacent to the correct target, reflecting the numerical distance effect. Performance tuning curves derived from the classifier's accuracy (Fig. 5A) further illustrate both numerical size and distance

effects, consistent with those observed in the monkeys' behavior (Fig. 2C, D) and neuronal population tuning (Fig. 4A). Decoding accuracy was substantially higher when using only numerosity-selective neurons—particularly for higher numerosities (Supplementary Fig. S3A)—than when using numerosity-unselective neurons (Supplementary Fig. S3B), demonstrating that number-selective cells make a critical contribution to population-level decoding of target numerosity.

Next, we tested whether the population code was independent of instruction stimulus format (dots vs. signs) by performing within-format classification. Classifiers were trained on one format (dots or

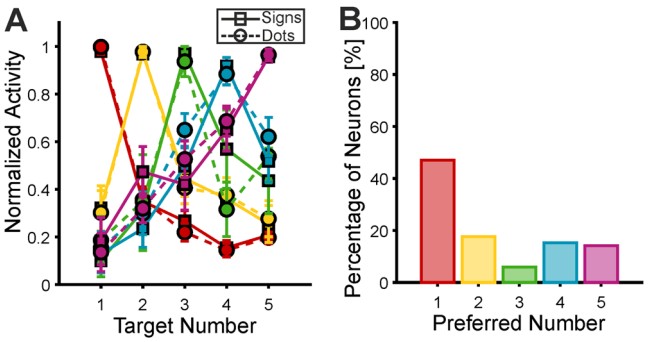

**Fig. 4 | VIP neurons are tuned to the impending number of hand movements.**
**A** Neuronal population tuning function during the motor planning period. The tuning functions of individual neurons were normalized and averaged according to their preferred numerical value (color-coded) for both stimulus formats. Error bars represent the SEM across neurons ($n = 40, 15, 5, 13, 12$ neurons for preferred numerical values 1–5, respectively). **B** Frequency of selective neurons preferring each of the five impending numbers of hand movements (1 to 5). (Source data are provided as a Source Data file).

signs) and tested on new trials from the same format. Classification accuracy was well above chance for both formats (dots: $58.6 \pm 0.2\%$; signs: $69.4 \pm 0.2\%$) (Fig. 5B). Lastly, we performed across-format classification to test whether classifiers trained on one cue format could decode target numbers in the other. For example, a classifier trained on dot trials was tested on sign trials, and vice versa. We found stable across-format accuracy in both directions (dots to signs: $60.3 \pm 0.2\%$; signs to dots: $58.2 \pm 0.2\%$) (Fig. 5B). This indicates a stable sensorimotor population code in VIP that encodes planned action numbers independent of stimulus format.

### Temporal evolution of sensory-to-motor transition of number information

Next, we examined whether a transformation of sensory numerosity representations into a motor planning code occurs within VIP. If such a transformation takes place, numerical information encoded during the sensory instruction phase should be preserved and transferable to the motor preparation phase. To this end, we investigated the continuity of numerosity coding across the instruction and motor planning phases.

Figure 6A shows that a classifier trained on the population activity of VIP neurons can decode the instructed number. Decoding emerges in the second half of the instruction stimulus phase and increases throughout the motor planning period. Importantly, decoding continues without interruption between the instruction and planning phases. This smooth transition from instruction to motor planning is further supported by the classifier tuning curves across time windows. As shown in Fig. 6A, B, decoding of the instructed—and impending—number increases continuously until the end of the planning phase.

In addition, we performed an across-phase classifier decoding analysis using the population of number-selective neurons. This analysis revealed that numerical information present during the instruction phase carried over into the motor preparation phase, and vice versa (Fig. 6C). Chance level was 20%, reflecting the five-number task. A classifier trained on the instruction phase achieved an accuracy of 47% when tested within the same phase (within-phase decoding) and 32% when tested on the subsequent motor preparation phase (across-phase decoding), which was significantly above the 95% confidence threshold of 26%. Conversely, a classifier trained on the motor preparation phase reached an accuracy of 68% when tested within the same phase and 34% when tested on the preceding instruction phase,

again significantly exceeding the 95% confidence threshold. These results demonstrate that numerical information encoded during the sensory instruction phase is preserved and transferable to the motor preparation phase, and vice versa (Fig. 6C).

Together, these findings suggest that VIP neurons do more than simply reflect a motor plan inherited from other areas. Rather, sensory numerosity information conveyed by the instruction stimulus is transformed into a preparatory motor signal representing the impending number of hand movements within VIP.

### Behavioral relevance of motor planning activity for the motor counting process

If the activity of tuned neurons predicted the monkeys' produced number of hand movements, neuronal responses during correct trials should differ compared to those during errors, where the monkeys performed fewer or more actions than instructed. Indeed, neuronal activity at the preferred target number was significantly reduced during incorrect trials compared to correct trials (100% vs. $79.3 \pm 3.2\%$; $14.7 \pm 2$ Hz vs. $12.5 \pm 1.8$ Hz; $n = 52$; $p = 1.1 * 10^{-7}$, Wilcoxon signed-rank test) (Fig. 7A). Moreover, neuronal activity during trials in which the tuned neurons' preferred number $n$ was produced erroneously (instead of the instructed $n-1$ or $n+1$) was higher than during trials where $n-1$ or $n+1$ was correctly produced (correct ($n-1$) vs. incorrect ($n$): $70.3 \pm 5.5\%$ vs. $102.1 \pm 9.2\%$, $p = 0.004$; correct ($n+1$) vs. incorrect ($n$): $78.2 \pm 3.5\%$ vs. $86.4 \pm 6.2\%$, $p = 0.084$; $n = 17$; Wilcoxon signed-rank test) (Fig. 7B).

If the sensorimotor population code is behaviorally relevant, classifier accuracy should reflect the monkeys' number production errors. Specifically, accuracy for correct trials is expected to decrease when the classifier is tested on error trials. Indeed, classifiers trained on correct trials showed significantly reduced accuracy when tested on error trials compared to correct trials (correct: $56.4 \pm 0.2\%$; error: $44.1 \pm 0.3\%$; Wilcoxon signed-rank test, $p = 3,98 * 10^{-130}$; $n = 1000$ resamples; $n = 129$ neurons) (Fig. 7C).

Next, we examined whether the population code could predict not only correct versus incorrect trials but also the direction of numerical errors—specifically, whether monkeys performed one fewer ($-1$ error) or one more ($+1$ error) hand movement than instructed. Remarkably, the classifier reliably distinguished between underestimation ($-1$ error) and overestimation ($+1$ error) trials of the correct number (all $p < 0.001$, Wilcoxon signed-rank test, $n = 1000$ resamples). Classifiers trained to distinguish these outcomes achieved the highest accuracy in predicting actual trial types: correct ($55.1 \pm 0.3\%$), $+1$ error ($55.1 \pm 0.3\%$), and $-1$ error ($47.2 \pm 0.3\%$) ($n = 157$ neurons) (Fig. 7D). Together, these findings indicate that the sensorimotor population code in VIP reliably represents perceived numerosity and encodes both planned actions and the outcomes of number production, including over- or underestimation errors.

### Static and dynamic codes underlying the sensorimotor translation

We then investigated the temporal dynamics and neuronal coding underlying the sensorimotor translation process using two time-resolved population analyzes. First, we performed an $\omega^2$ percentage of explained variance (PEV) analysis on the population activity. The PEV quantifies the proportion of neural activity variance explained by specific task-related factors, reflecting how strongly these factors are encoded in the neuronal population. We calculated the $\omega^2$ PEV for numerosity, stimulus format (dots vs. signs), and their interaction. The neuronal population encoded numerosity significantly, while neither stimulus format nor the interaction between the two showed significant encoding (Fig. 8A). Neural encoding of numerical information emerged late in the instruction stimulus phase and steadily increased during motor planning, reaching its peak at the onset of the response phase. This temporal profile indicates an abstract, population-level

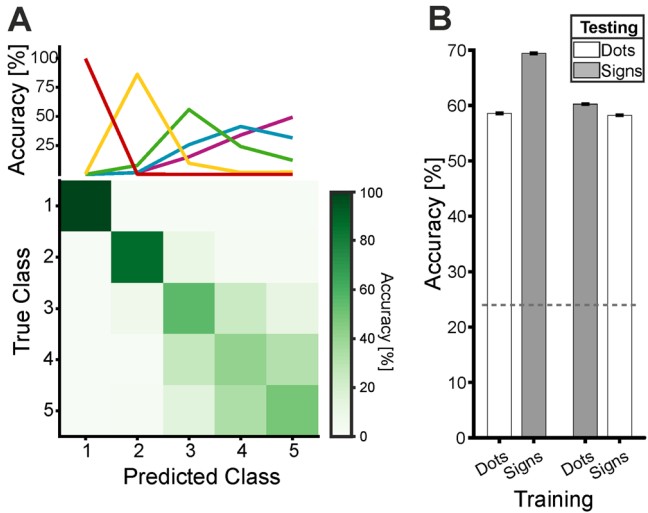

**Fig. 5 | Population of VIP neurons represents the impending number of hand movements. A** Performance of an SVM classifier decoding target numerosities (1 to 5) from the firing rates of all VIP neurons during the motor planning period. *Top*: Accuracy curves for each target numerosity, color-coded. *Bottom*: Corresponding confusion matrix showing predicted versus instructed number of movements. The main diagonal indicates correct classifications. **B** Classification accuracy within and across formats. *Left*: Within-format accuracy (training and testing on the same format). *Right*: Across-format accuracy (training and testing on different formats). The dotted line indicates the 95th percentile of accuracies from shuffled labels. Error bars represent SEM (averaged across 10-fold cross-validation and 1000 resamples, $n = 246$ neurons). (Source data are provided as a Source Data file).

representation of numerosity in VIP neurons during the sensorimotor transformation from perception to action.

We further examined how numerosity encoding dynamically evolves into a representation of the planned number of hand movements using cross-temporal classifier analysis. Here, SVM classifiers were trained on neuronal firing rates at one time point and tested on firing rates from different time points in separate trials. The resulting accuracy was visualized in a matrix comparing training and testing times across the trial, with diagonal values—where training and testing occur at the same time points (Fig. 8B). Classifier accuracy remained significantly above chance throughout both the instruction stimulus and motor planning periods, demonstrating continuous transition from sensory numerosity representation into a motor planning code for the upcoming number of hand movements (Fig. 6C).

Two extreme coding schemes can be envisioned for the sensorimotor transformation process. In a static code, neurons maintain persistent tuning to a specific target number throughout the sensory instruction and motor planning periods, allowing a decoder trained at one time point to generalize across others—resulting in high decoding accuracy extending beyond the diagonal of the cross-temporal matrix[22]. Conversely, a dynamic code features neurons with transient, rapidly shifting tuning over time, such that decoding accuracy is high only when training and testing occur at the same time point, producing strong performance confined to the diagonal[23].

We found evidence for both coding schemes. First, a static code was supported by significant cross-temporal generalization extending from the onset of the instruction stimulus through the motor planning phase (Fig. 8B). Classifiers trained on firing rates during the instruction period reliably decoded the planned number of actions when tested on activity recorded later in the motor planning period, producing a distinctive square-like pattern of above-chance accuracy in the cross-temporal matrix (highlighted by thick contours in Fig. 8B). A cluster permutation test confirmed this pattern's significance (see

"Methods"), indicating that numerosity information generalized across these trial phases. This partial generalization aligns with the presence of a static neuronal code in VIP, consistent with individual neurons maintaining stable tuning to the target number throughout both instruction and motor planning intervals (Fig. 8C).

In addition, we observed signatures of a dynamic code, characterized by high classification accuracy narrowly confined to the main diagonal of the cross-temporal matrix (Fig. 8B). Classifiers trained on firing rates from specific time windows after instruction onset performed well only when tested on the same or nearby time points, with a brief dip in accuracy at the motor planning onset and a resurgence toward the end of the planning phase. This limited temporal generalization indicates that numerosity encoding in VIP also involves rapidly shifting neuronal patterns, consistent with individual neurons exhibiting transient tuning restricted to brief trial intervals (Fig. 3E). A mixture of both static and dynamic coding was also present for neurons decoding the specific number (Supplementary Fig. S4). The increase in decoding accuracy toward the execution phase is not driven specifically by numerosity 1 trials. Together, these findings suggest that VIP employs a hybrid coding scheme, combining stable and dynamic representations during sensorimotor number transformation.

## Discussion

We show that monkeys can translate varying numerical instructions into the corresponding number of self-generated hand movements and actively signal the end of their motor counting sequence—an especially demanding cognitive task. In doing so, they exhibit hallmark features of the ANS, such as bell-shaped performance curves reflecting a numerical distance effect consistent with Weber's law[2]. The monkeys demonstrated remarkable precision in estimating the correct number of actions, comparable to humans performing similar tasks through rapid key presses without relying on symbolic counting strategies[10,11]. Both monkeys performed better with sign formats than with dot numerosities[24]. We suspect that this difference reflects the higher visual variability and perceptual noise inherent in dot numerosity displays, whereas the sign stimuli serve as clear instruction cues that signal numerical information more reliable.

During performance of this task, VIP neurons were tuned not only to the instructed numerical value[25] but also predicted the number of upcoming self-initiated hand movements in a behaviorally meaningful way. Population decoding analyzes showed that VIP reliably represents perceived numerosity and encodes planned actions. Moreover, this population code reflected the outcomes of number production, including over- and underestimation errors.

Numerical selectivity in VIP during the motor preparation period reflects more than a mere representation of a learned motor plan. Number selectivity emerges already during the sensory instruction phase, persists across task phases, and generalizes across sensory and motor periods, as shown by within- and across-phase decoding and time-resolved population analyzes. The uninterrupted, continuous decoding of number information from instruction through motor preparation supports the notion that VIP transforms sensory numerosity representations into a motor planning code, rather than simply reflecting a motor-specific signal detached from sensory and categorical input. This interpretation is consistent with previous neuron recordings from monkey parietal association cortex. They show that IPS neurons encode abstract, categorical decision variables that bridge sensory input and action planning rather than representing sensory or motor information in isolation[26–29]. Similar to reports from LIP and MIP, where decision- and category-related signals are progressively reflected in motor-preparatory activity, our results suggest that VIP transforms abstract numerical information into task-appropriate motor plans. This process involves actively maintaining numerical cues in working memory and gradually converting them into numerical intent

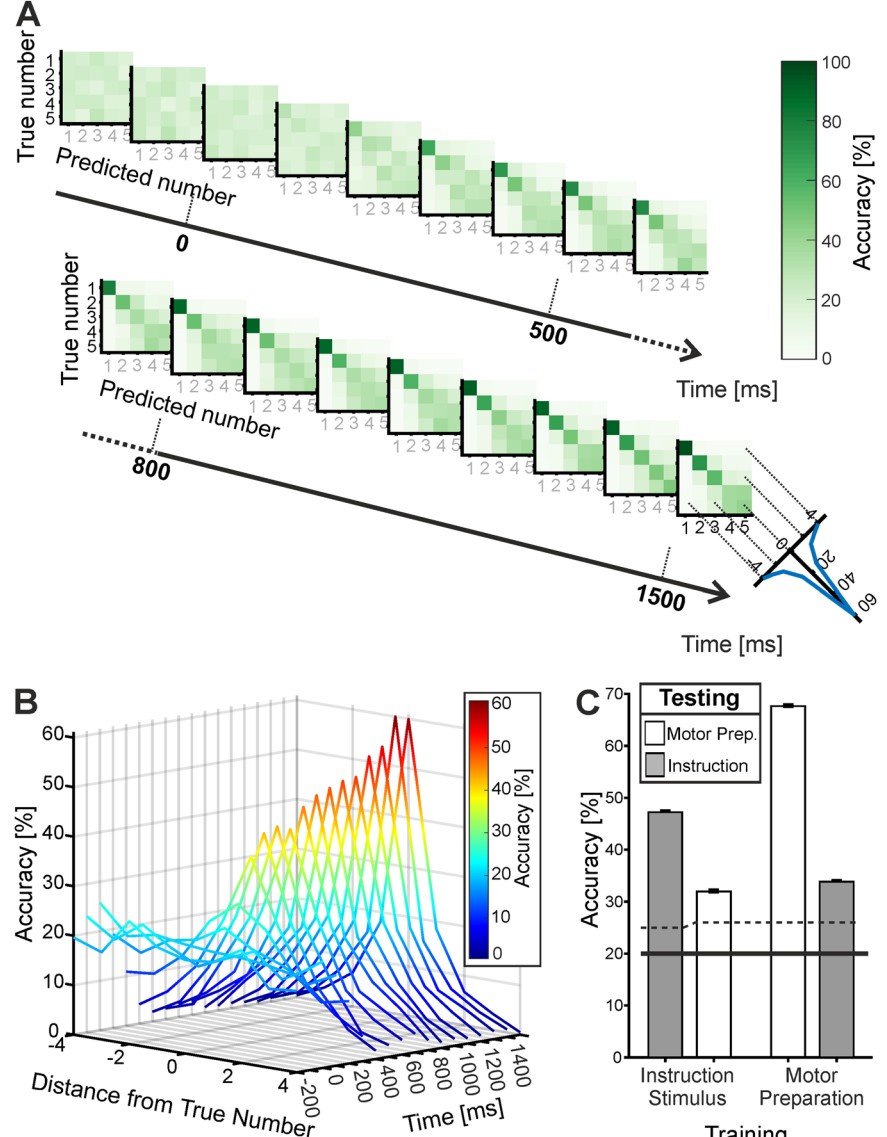

**Fig. 6 | Temporal evolution of sensory-to-motor population decoding of number information. A** Confusion matrices showing the performance of SVM classifiers decoding numerosity (1–5) from VIP population firing-rate activity (200 ms window, 100 ms step size) across the fixation, instruction, and motor preparation phases (−300 to 1600 ms relative to sample onset). Classifiers were trained and tested within the same time intervals. Each confusion matrix depicts the accuracy of predicted relative to instructed numerosities; the main diagonal indicates correct classifications. The tuning curve in the bottom right panel shows the average classification probability for each numerosity (correct label centered) across trial phases averaged across 10-fold cross-validation and 100 samples, for 246 neurons. **B** Depiction of classifier tuning curves for each time window shown in **A** across the fixation, instruction, and motor preparation phases. Classifier tuning curves were derived from classification probabilities as a function of distance from the correct label over time. **C** Within- and across-phase decoding for the instruction stimulus and motor preparation phases. Left: average classifier accuracy for a classifier trained on number during the instruction stimulus phase (300 ms interval, starting 250 ms after instruction onset) and tested either within the instruction phase or across phases during the motor preparation phase (900 ms interval, starting 200 ms after motor preparation onset). Importantly, the analysis windows are non-overlapping. Right: average classifier accuracy for a classifier trained during the motor preparation phase and tested either within the motor preparation phase or across phases during the instruction stimulus phase. The dotted horizontal line indicates the upper 95% confidence threshold; the solid line indicates chance level (20%). Error bars represent SEM (averaged across 10-fold cross-validation and 1000 resamples, $n = 93$ neurons). (Source data are provided as a Source Data file).

before execution. Together, these findings indicate that VIP, along with other IPS areas[12], functions as a sensorimotor bridge linking perceived numerosity to impending action[16], rather than merely reflecting a passive motor plan.

Our cross-temporal population decoding analysis further illuminates the nature of neuronal coding during numerical sensorimotor transformation. They suggest that in monkey VIP, both sustained neuronal tuning (static code) and transient tuning (dynamic code) coexist during the process of converting sensory number signals into quantitative motor plans, likely playing complementary roles[22,23]. This

dual coding is common in the primate cortex, where persistent, stable activity with strong across-time generalization coexists alongside dynamically evolving neuronal patterns[30,31]. Static codes may support retrospective working memory by maintaining past information[32], whereas dynamic codes may guide prospective processes, preparing future actions[33]. Since our monkeys converted a remembered numerical cue into a planned sequence of hand movements, these overlapping codes likely represent the retention and anticipation phases of this transformation. More broadly, such multiplexed coding may be a general neural mechanism for converting sensory inputs into

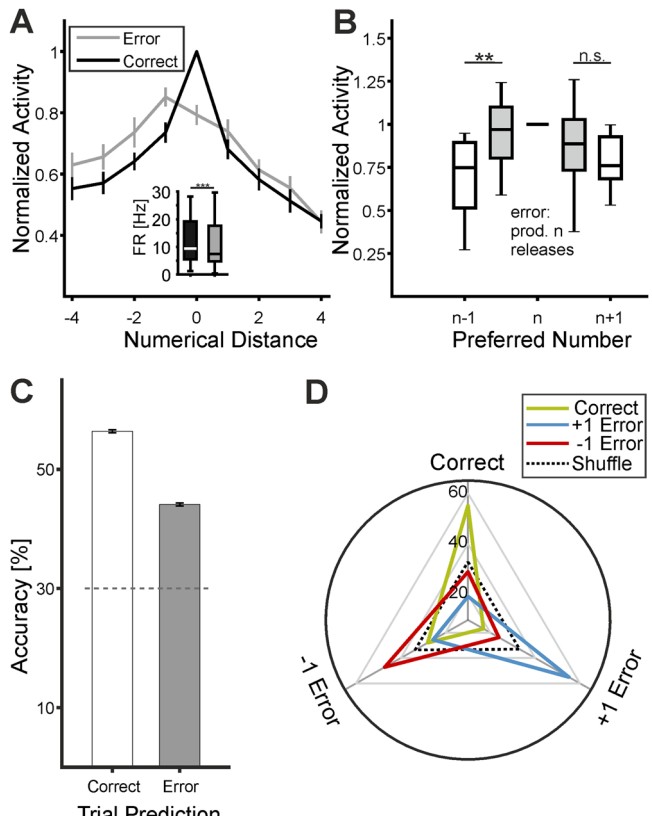

**Fig. 7 | Behavioral relevance of VIP activity based on correct versus error trials.**
**A** Average neuronal population tuning curves plotted as a function of numerical distance from the respective preferred numerical values (i.e., distance 0). Tuning for the same neurons is shown during both correct and incorrect trials. Error bars represent the SEM where applicable ($n = 52$ neurons). The inset displays the average firing rate to the preferred number during correct and incorrect trials as boxplots ($n = 52$ neurons, *** $p = 1.1*10^{-7}$, Wilcoxon signed-rank test). The median is represented by the line within each box. The box spans the interquartile range from the 25th to the 75th percentile. The whiskers reach from the minimum to the maximum. **B** Average normalized activity in trials in which a neuron's preferred target number $n$ was produced as an error (gray boxes) instead of instructed $n + 1$ or $n − 1$ handle releases. The average activity during correct trials is depicted in the white boxes. The box spans the interquartile range from the 25th to the 75th percentile, and the whiskers reach from the minimum to the maximum. The line within each box indicates the median. ($n = 17$, ** $p = 0.004$; n.s. $p = 0.084$; Wilcoxon signed-rank test). **C** Accuracy of SVM classifiers trained on correct trials during the motor planning period to predict the number of movements on either correct or incorrect trials. The dotted line shows the 95th percentile of shuffled label accuracies. Error bars represent SEM (averaged across 10-fold cross-validation and 1000 resamples, $n = 246$ neurons). **D** SVM classifier performance in predicting whether monkeys made the correct number of movements, one more (+1 error), or one less (−1 error) than instructed. The classifier successfully decoded both correct outcomes and the direction of errors from activity during the motor planning period. The dotted line indicates chance level (33%) (averaged across 10-fold cross-validation and 1000 resamples, $n = 246$ neurons). (Source data are provided as a Source Data file).

volitional, goal-directed motor outputs. Understanding how the brain encodes and converts abstract quantities into actions could aid the development of brain-machine interfaces for neuroprosthetics that interpret numerical intentions and translate them into precise motor commands[34].

Consistent with findings in macaques, where neurons in area VIP respond to both visual and auditory numerosity[5–8], human homologs of VIP and LIP have also been shown to encode the number of items in visual arrays[35]. Notably, a nearby intraparietal region just anterior to

VIP−known as AIP−is activated during numerical processing of observed actions in humans[36]. In addition to its role in numerosity, the human VIP shares functional similarities with its macaque counterpart by integrating multiple sensory inputs−visual, tactile, and vestibular− to support near-face defense, motion detection, and the coordination of hand-to-mouth movements[37,38]. The close anatomical and functional links between parietal regions involved in both hand movement and numerical representation may help explain how perceived numerical information is translated into self-generated hand actions. In conjunction with the prefrontal cortex−which is crucial for maintaining working memory of supramodal numerosity in both monkeys[8] and humans[39,40]−and with mediotemporal regions that house numerosity-selective neurons[41,42], these parietal areas likely form the core of a sensorimotor number system in the human brain[16]. This system may not only support nonsymbolic numerical transformations in non-human primates but also symbolic number processing underlying human mathematical reasoning.

The idea of a sensorimotor number transformation process aligns with evolutionary and comparative evidence suggesting that numerical representations reflect innate computational properties of the primate brain rather than being solely a byproduct of training. Numerosity perception and action planning appear tightly linked in a sensorimotor numerosity system that integrates sensory input with motor outcomes across species, indicating that numerical processing serves both perception and action[16]. From an evolutionary perspective, many animals, including non-primates, can translate perceived numerosity into corresponding actions[43–45], suggesting that basic sensorimotor representation is ancient and widespread, likely supporting adaptive functions such as action planning and communication[46–49]. Psychophysical work in humans further shows that motor actions, such as tapping, influence perceived numerosity, consistent with shared sensorimotor channels for numerical and motor information[50]. Thus, the transformation we observe in VIP likely reflects an intrinsic tendency of primate neural circuits to link abstract quantity with action plans, rather than a product of laboratory reinforcement learning. The IPS appears to be part of a conserved network that supports numerical estimation and its translation into behaviorally relevant actions across contexts.

## Methods
### Subject details
Two adult male rhesus monkeys (*M. mulatta*), aged 8 and 7 years, served as subjects in this experiment. During the study, they had average body weights of 11.5 kg and 7.8 kg, respectively. The monkeys were seated in a primate chair within a darkened chamber during task performance. Visual stimuli were presented on a 15-inch flat-screen monitor (1024×768 resolution at 75 Hz) positioned 57 cm in front of the animals. Correct trial completion was rewarded with fluid, delivered via a computer-controlled valve. All experimental procedures conformed to the guidelines for animal experimentation and were approved by the national authority (Regierungspräsidium Tübingen, Germany).

### Behavioral protocol
We trained two rhesus monkeys to plan and produce a specific number of identical motor responses−handle releases−in response to visually presented numerical cues. Task control and behavioral monitoring were implemented using CORTEX software (NIMH, Bethesda, MD). To initiate a trial, the monkey fixated on a central square and gripped a handle. Eye position was continuously monitored with an infrared eye tracker (ISCAN ETL-200, ISCAN Inc., Woburn, MA). An instruction stimulus (500 ms) indicated the number (1−5) of handle releases to be executed, which the monkey held in working memory during a 1000 ms motor planning period.

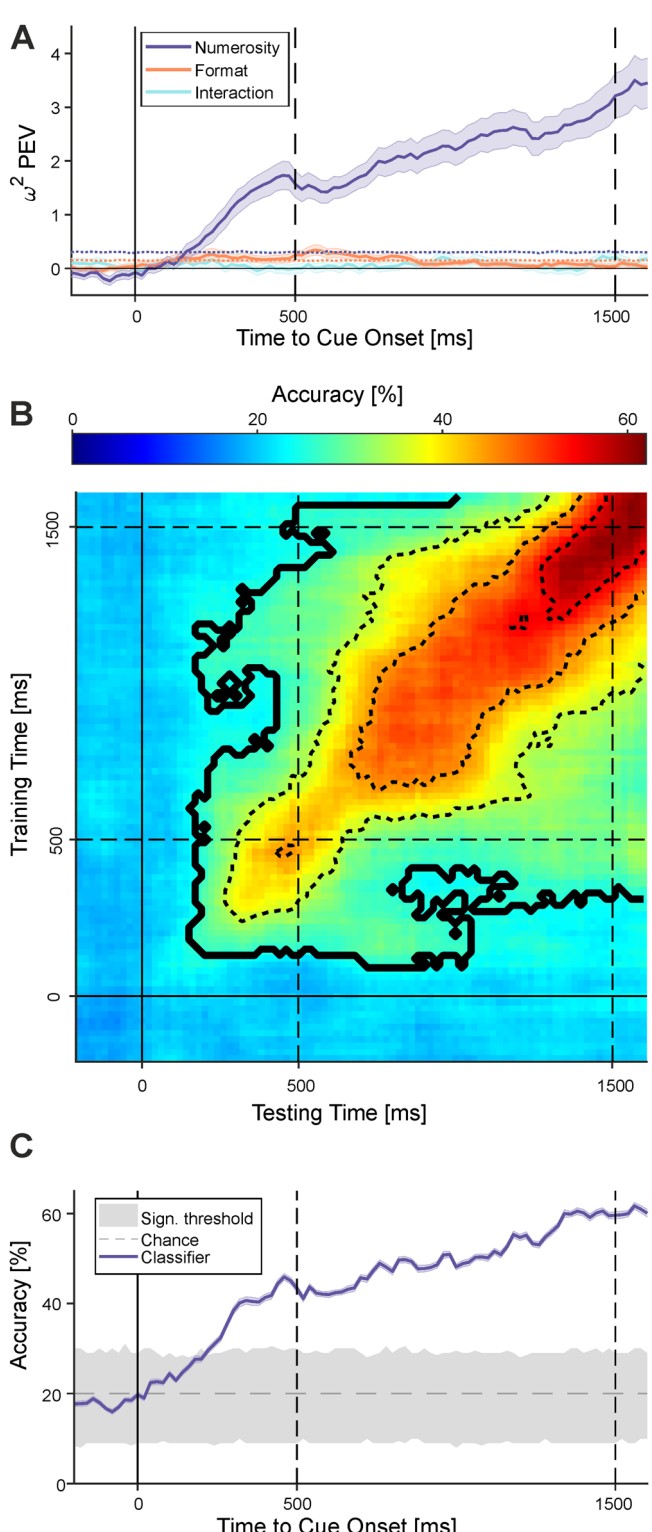

**Fig. 8 | Neural codes during temporal transformation in the population of VIP neurons. A** Encoding of task-related information—target numerosity, instruction stimulus, and their interaction—quantified as percentage of explained variance (PEV) over the course of a trial. Time 0 ms marks the onset of the instruction stimulus. Shaded areas represent SEM across resamples. Dotted lines indicate the PEV from shuffled trial labels. **B** Cross-temporal classification accuracy. Mean accuracy is color-coded in a 2D matrix, with training times on the *x*-axis and testing times on the *y*-axis. Time 0 ms marks instruction stimulus onset (solid black line), and 500 ms marks the start of the motor planning period (dashed line), which lasts until 1500 ms. Significant accuracy clusters at 25% accuracy are outlined in black (cluster permutation test). Dashed contour lines indicate accuracy levels from 35% to 65% in steps of 10%. **C** Mean decoding accuracy of the time-resolved SVM classifier shown in b, plotted over the time course of the trial. The gray shaded area represents the distribution of shuffled-label classifier performance (5th to 95th percentile), and the dashed line indicates chance level (20%) for decoding five labels. (Source data are provided as a Source Data file).

signaled completion by shifting gaze to the confirmation stimulus (within ±3.5°). A correct trial, in which the number of releases matched the instructed number, was rewarded. Importantly, the temporal arrangement of the motor execution period was unknown to the monkeys and could not have influenced neuronal activity during motor planning.

Error trials (early confirmation or overshooting the instructed number) were marked by a red screen for 1 s and were unrewarded. Gaze deviation >3.5° or premature bar release before enumeration cue onset resulted in immediate trial abortion, indicated by a brief (1 s) blue or green screen, but not classified as an error. In such cases, the same trial was repeated. Two presentation protocols (standard and control conditions) using the numerals 1–5 were presented in a pseudorandom, balanced sequence.

### Training procedure

Animals were trained using positive reinforcement, with water delivered as a reward for correct trials. Both subjects were numerically naïve at the start and had previously been trained only on a delayed match-to-color task. Training progressed in individually adjusted steps based on each animal's performance. Starting from the delayed match-to-color task, sign stimuli representing numerosities one and two were introduced first. Higher numerosities (three to five) were then added sequentially, initially using color-coded cues. Once these steps were successfully acquired, the temporal structure of the task was gradually modified: waiting periods were progressively extended until the animals reliably waited for the enumeration cue before releasing the handle. Dot-format stimuli were then introduced in a stepwise manner, beginning with numerosities one and two and then sequentially adding three to five. In the final training phase, both stimulus formats (sign and dot) and all temporal arrangements were combined to complete the task.

### Stimuli

To present numerical values, we used two distinct stimulus regimes representing the numbers 1 to 5. In the dot format, instruction stimuli consisted of arrays of 1 to 5 black dots displayed on a gray circular background. In the sign format, the stimuli were white Arabic numerals (1–5) that the monkeys had been trained to associate with the corresponding number of handle releases.

To control for non-numerical visual features, each protocol included a standard and a control condition. In the dot format's standard condition, dot size and location were pseudo-randomized while ensuring that dots did not overlap or touch. The control condition matched stimuli for low-level visual attributes such as total area and dot density. For the sign format, the standard condition used numerals presented in "Arial" font, while the control condition used

The motor execution phase began with the appearance of an enumeration cue: a grey square located 6° above the fixation point. A background circle, present throughout the planning period, now served to contextualize the numerical response. Each handle release had to occur within 500 ms of cue appearance. After each release, the cue temporarily disappeared before reappearing to prompt the next response. Three different temporal arrangements of handle releases, controlling for total release time and rhythmicity, were pseudorandomly predetermined to prevent the monkeys from using timing cues (Fig. 1C). Upon completing the planned sequence, the monkey

"Times New Roman" to introduce perceptual variation without altering numerical meaning. All stimuli were presented centrally on a uniform gray background and generated using MATLAB (MathWorks Inc., Natick, MA) prior to each session.

## Temporal arrangement during the enumeration period

As noted above, the intervals between responses during the motor execution period were variable. These intervals were systematically manipulated to prevent the monkeys from adopting temporal or rhythmic strategies to solve the task. Three temporal arrangements were used: one standard and two control conditions.

In the standard timing arrangement, the inter-response interval was pseudo-randomly selected from four predefined durations (200 ms, 500 ms, 800 ms, and 1.1 s), each with equal probability. This prevented the emergence of rhythmic patterns.

The first control arrangement implemented a fixed wait interval, in which all inter-response intervals were set to a constant 200 ms. The second control arrangement used a fixed overall duration, in which the total length of each trial was held constant across numerosity values. Based on an estimated average reaction time of 300 ms, the wait intervals were adjusted accordingly: for numerosity 2, a single interval of 2.1 s; for numerosity 3, two intervals of 800 ms; for numerosity 4, three intervals of 400 ms; and for numerosity 5, four intervals of 200 ms.

In each session, the standard timing condition was presented alongside one of the two control arrangements in a pseudo-randomized, balanced fashion. The control timing conditions alternated daily.

## Surgery and recordings

All surgical procedures were performed under general anesthesia and aseptic conditions. Prior to behavioral training, both monkeys were surgically implanted with a titanium head post to allow for stable head fixation during recording sessions and reliable monitoring of eye movements. After the completion of behavioral training, each monkey received a recording chamber implant targeting the IPS, positioned based on individual anatomical MRI scans and stereotactic coordinates. The chamber was implanted over the left hemisphere in both animals.

Electrophysiological recordings were conducted using an array of eight glass-coated tungsten microelectrodes (Alpha-Omega Engineering, Israel) with an impedance of ~1 MΩ. The recording sites were determined based on individual a priori MRI scans from both monkeys, combined with reconstructed coordinates and electrode track depths. Electrodes were inserted transdurally from the cortical surface to depths of 8–12 mm, each attached to a custom-made mechanical microdrive. Entry points were located on both the dorsal and ventral banks to access VIP in the fundus of the IPS. In previous experiments[7], we found global visual motion direction selectivity and tactile responses at the target recording sites, consistent with the known functional properties of VIP. Neural signals were amplified, bandpass filtered, and digitized at 40 kHz using a Plexon MAP system (Plexon Inc., Dallas, TX) for offline spike sorting.

## Data analysis

All data analyzes were performed using MATLAB (MathWorks Inc., Natick, MA). Unless otherwise stated, all values presented in the figures and main text are reported as mean ± standard error of the mean (SEM). The SEM was calculated by dividing the standard deviation by the square root of the sample size.

## Behavioral analysis

Overall performance (in %) was calculated for each session as the number of correct trials divided by the total number of correct and incorrect trials. To estimate the chance level, we analyzed the distribution of unrestricted responses across sessions and identified the most frequently chosen number, which was 8. Based on this, the chance level was set at 12.5%. Sessions with overall performance below 40% were excluded from analysis, as they likely reflected insufficient task engagement or motivation. Behavioral performance curves were generated by calculating the relative frequency of responses for each instructed numerosity, separately for the dot and sign formats, and then averaged across recording sessions.

## Neuronal tuning and selectivity analysis

For the following analyzes, single neurons were included if they exhibited a mean firing rate of at least 0.5 Hz during the relevant task period, defined as the 2000 ms interval from the onset of the baseline to the end of the motor planning phase, and if at least five correct trials were available per condition. We defined 20 distinct trial conditions, encompassing five numerical values (1–5) across two stimulus formats (signs, dots) and two stimulus conditions (standard, control).

To identify numerosity-selective neurons during the instruction and motor preparation phases, we performed a two-factor sliding window ANOVA (window size: 200 ms; step size: 20 ms; significance threshold: $\alpha < 0.01$) starting 100 ms after instruction stimulus onset and extending 100 ms into the motor execution phase. The two factors were "numerosity" (1–5) and format (signs vs. dots). The "stimulus condition" (standard/control) was not included due to differences in visual structure across formats. Thus, in both the instruction stimulus phase and the motor preparation phase, neurons were characterized based on numerical values, i.e., abstract numerical categories. During the instruction stimulus phase, these categories were cued by dot displays and associated signs. During the motor preparation phase, they corresponded to the instructed number of hand movements.

We additionally analyzed neural activity during the sample period using a two-factor sliding-window ANOVA to identify numerosity-selective neurons. The analysis was performed in 500 ms windows, starting 100 ms after stimulus onset and extending 100 ms into the motor planning phase. We also quantified the proportion of neurons exhibiting main effects of numerosity, stimulus format, or their interaction across both task periods.

A neuron was classified as numerosity-selective if it showed a significant main effect of numerosity without a main effect of format or an interaction between the two factors for at least 11 consecutive time bins (≥300 ms total duration). If multiple selective intervals were found, the one with the largest firing rate difference between numerical values was chosen for further analysis. Each neuron's preferred numerosity was defined as the numerical value eliciting the highest average firing rate within its selective window. To construct population response functions, firing rates were normalized within each neuron to its minimum and maximum response and then averaged across neurons grouped by their preferred numerosity.

To validate this method, we performed split-half cross-validation of tuning curves. Trials for each neuron were split into two balanced halves (50/50). A tuning function and preferred value (maximum response) were determined from one half, and responses from the other half were normalized accordingly. Next, the tuning function and preferred value were independently estimated from the second set, and the similarity between tuning functions from both sets of data was quantified using Pearson correlation. Significance was assessed by permutation of stimulus labels to generate pseudo-tuning functions and recompute a distribution of correlation values 10,000 times. The percentile position of the observed correlation within this permutation distribution was taken as a measure of tuning reliability.

We assessed how many numerosity-selective neurons showed excitatory versus inhibitory responses by comparing the average firing rate during the numerosity-selective interval (collapsed across all five numerosities) with baseline activity during the fixation period.

Neurons were classified as excited if their activity exceeded baseline and as inhibited if it fell below baseline.

### Error trial analysis

To assess the behavioral relevance of motor planning activity, we compared neuronal responses between correct trials and those in which the monkey produced an incorrect number of movements. Specifically, we focused on each neuron's preferred numerosity and compared its activity during correct versus error trials using a Wilcoxon signed-rank test. For inclusion, neurons required a minimum of three incorrect trials ($n = 52$). Population response functions were constructed by averaging normalized firing rates (across both correct and error trials) and aligning each neuron's preferred numerosity to position zero.

In a subsequent analysis, we examined neuronal activity in trials where the neuron's preferred numerosity was produced erroneously— i.e., the monkey executed the preferred number of actions ($n$) instead of the instructed number ($n+1$ or $n-1$). This analysis was limited to neurons for which at least three such incorrect trials were recorded ($n = 17$). Due to the proximity requirement for misclassification, only neurons tuned to the numerosity values 2, 3, or 4 were included.

### Classifier analyses

Classification analyses were conducted using a linear multi-class SVM model with one-vs.-one encoding to address the five-class problem[51]. All analyzes were performed on trial-averaged firing rates computed during the 900 ms motor planning period. For overall population classification, neurons with at least 20 correct trials per numerosity were included. From each included neuron, 20 trials per class (numerical values 1–5) were randomly sampled (totaling 100 trials), z-scored, and subjected to a tenfold cross-validation procedure. In each iteration, 90 trials were used to train the SVM and the remaining 10 for testing. This process was repeated 10 times with different trial partitions, and the full procedure was iterated 1000 times with different random subsets to ensure robust accuracy estimates. Chance-level performance was assessed by repeating the procedure on datasets with shuffled class labels.

Confusion matrices were generated by comparing predicted and actual labels, and overall classification accuracy was quantified as the mean of the diagonal entries (i.e., correct classifications) of these matrices.

To test cross-format information transfer between the two stimulus formats (dots and signs), SVM models were trained on one format and tested on the other. For this analysis, neurons with at least 10 trials per numerosity per format were included. Using a tenfold cross-validation design, nine trials were used to train the classifier and one for testing, performed both within-format and across-format.

To assess behavioral relevance, we tested classification performance on error trials. Neurons with at least one incorrect trial per class were included (noting fewer errors for numerosity 1). Classifiers were trained on nine correct trials and tested on one correct trial plus one randomly selected error trial. This was performed using tenfold cross-validation and repeated 1000 times.

Lastly, to decode trial outcomes directly from population activity, we defined three classes: correct responses, −1 errors (undershoots), and +1 errors (overshoots). Only neurons with at least 30 trials per class were included. The data were split into 10 folds, using 27 trials for training and three for testing, and accuracy was evaluated similarly as above.

### Time-resolved population analyses

The percent explained variance (PEV) is a widely used measure to quantify the amount of information carried by neuronal populations over time[52]. In this study, we evaluated how much information neurons conveyed about two task-relevant factors: numerical value and

stimulus protocol. Only neurons with at least 10 correct trials per factor level (numerical value and stimulus protocol) were included.

To capture time-resolved effects, we applied a two-factor sliding window ANOVA with a window size of 200 ms and a step size of 20 ms. This analysis was conducted over a temporal window starting 300 ms before instruction stimulus onset (baseline) and extending to 100 ms after the motor planning period ended. The ANOVA yielded sums-of-squares values for each factor, which were then used to compute the variability measure $\omega^2$ (omega squared) as a function of time according to:

$$\omega^2 = \frac{SS_{TERM} - df \times MS_{ERROR}}{SS_{TOTAL} + MS_{ERROR}} \times 100 \qquad (1)$$

with $SS_{TERM}$ denoting the sum of squares for the term of interest (factors number, format or interaction), $SS_{TOTAL}$ being the total sum-of-squares, df are the degrees of freedom and $MS_{ERROR}$ referring to the mean squared error. Population activity was quantified by averaging the firing rates across neurons. To ensure robustness, we repeated the analysis 50 times using different randomly drawn trial subsets, calculating the mean and SEM across these repetitions. Baseline PEV was established by shuffling trial labels 20 times per resample, resulting in 1000 reshuffle values per time step (50 repetitions × 20 shuffles). The 95th percentile of this shuffled distribution was used to define the baseline PEV threshold.

To capture the temporal dynamics of sensory-to-motor transformation, we conducted a cross-temporal classification analysis using a linear multi-class SVM. Neurons with a minimum of 20 correct trials per target numerosity were included. Firing rates were binned with a sliding window of 200 ms and a step size of 20 ms. For each time window, an SVM model was trained using a tenfold cross-validation approach, with 18 trials for training and 2 trials for testing. Importantly, models trained on one time window were also tested on all other time windows, producing a two-dimensional matrix of decoding accuracy across training and testing times.

This entire procedure was repeated 50 times with different random trial samplings. Statistical significance was assessed using a cluster-based permutation test: during each cross-validation and redraw iteration, trial labels were shuffled 20 times, generating 1000 reshuffled accuracy values per time point to estimate chance performance. Clusters of contiguous significant time points ($\alpha_{cluster} = 0.05$) were identified based on whether neighboring time bins exceeded the threshold. Cluster sizes were compared against a null distribution from permuted data, and clusters with sizes exceeding the 95th percentile of this null distribution ($\alpha_{rank} = 0.05$) were deemed statistically significant.

To further characterize the transformation underlying number-specific decoding from the instruction phase to the onset of the motor planning phase, neural population activity was segmented into overlapping 200 ms time bins with a 100 ms step. Binning spanned from 300 ms before instruction stimulus onset to 100 ms into the execution phase, yielding a total analysis window of 1900 ms. For each time bin, population-level classification was performed using an SVM classifier, and confusion matrices were computed across 1000 repetitions. Decoding performance was quantified by averaging classification probabilities along diagonals parallel to the main diagonal, providing a measure of accuracy as a function of numerical distance from the correct label. To assess cross-phase information transfer between the instruction and motor preparation phases, SVM classifiers were trained on neural activity from one phase and tested on activity from the other. The instruction phase was defined as neural activity within a 300 ms interval starting 250 ms after instruction onset, and the motor preparation phase as neural activity within a 900 ms interval starting 700 ms after instruction onset. Only neurons with a minimum of 20 trials and selective to numerosity in either sample or motor planning

**Article** https://doi.org/10.1038/s41467-026-73037-9

phase were included in this analysis. Classification was performed using a tenfold cross-validation procedure, with 18 trials used for training and 2 trials for testing in each fold. Decoding accuracy was evaluated for both within-phase and cross-phase training–testing combinations.

### Reporting summary

Further information on research design is available in the Nature Portfolio Reporting Summary linked to this article.

## Data availability

Source data are provided with this paper Source data are provided with this paper.

## Code availability

The data and code that support the findings of this study are available from Figshare (https://doi.org/10.6084/m9.figshare.31096300).

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

## Acknowledgements
The authors would like to thank the animal care staff for their dedicated support and assistance throughout this study. This work was supported by a DFG grant NI 618/13-1 and NI 618/17-1 (FOR 5159) to A.N.

## Author contributions
A.N., S.W. and L.S. designed the experiment. L.S and S.W. conducted the experiments. L.S., S.W. and A.N. analyzed the data. L.S. and A.N. wrote the manuscript. L.S., S.W. and A.N. edited the manuscript. A.N. supervised the study.

## Funding

## Competing interests
The authors declare no competing interests.
