## [Transparent Peer Review file · Nature Communications]

Sensorimotor transformation of number in the primate parietal cortex

Corresponding Author: Professor Andreas Nieder

Version 0:

Reviewer comments:

Reviewer #1

(Remarks to the Author)

In this article by Seidler et al., the authors investigated the neural mechanisms for transformation of perceived numerical values into corresponding numbers of self-generated actions. For this purpose, they trained the monkeys to perform the task in which the subjects viewed visual numerical cues and produced a corresponding number of hand movements. They recorded the single neuron activities from VIP and found neurons tuned to the number of intended actions. Population decoding confirmed that these VIP neurons encoded intended number of actions including over- and underestimated errors. Thus, the neuronal mechanisms for conversion of abstract numerical input to actions were clarified for the VIP neurons. The study was well designed, and results are clear. But I have to make some comments about some unclear documentation to make this manuscript more concise to the readers.

Major comments:

1. Fig. 1C shows that a variety of intervals between each wait period were tested. The authors may wish to show that such variation did not affect the neural code of numerical values. But how was the result? At least, the authors should show some quantitative evidence that such manipulation did not affect the results.
2. As I see the example data in Fig. 3, the temporal profile of the neuronal responses in panels C-G are quite variant. There, the peak of the activity seems to be earlier for the neurons encoding smaller numbers. Is it true, or what the authors meant? Then, how would the panel B look like, if individual neurons in the panel are indicated with the colors corresponding to the best preferred number?
3. Static vs dynamic codes. The data in Fig. 6 shows that there are both static and dynamic codes. If these are dissociable, the readers may wonder how the static and dynamic codes appear for each neuron class with different preference for the number. I wonder whether the authors can show this kind of data.

Minor comments:

4. I feel some discomfort in calling different kind of format in Fig. 1B “standard” and “control”. They were prepared just for comparison to exclude the possible involvement of non-numerical factors. I feel more confirmable to label them “format 1” and “format 2” etc.
5. Rightmost part of the data in Fig, 3E and G might have been missing. Please show the full dataset.

Reviewer #2

(Remarks to the Author)

The goal of this paper is clear and straightforward: to test whether neurons in the VIP encode the transformation of visually perceived numerosity into the corresponding number of planned motor actions. The introduction situates the research question well within the existing literature, and the issue addressed is clearly novel. The manuscript is well written and pleasant to read. However,

I have a single major concern regarding the interpretation of the findings. In my view, the key sentence appears on page 4, line 61: “the monkeys translated the perceived numerical value into a planned sequence of corresponding hand movements”. I am not an expert in this specific field, but I would like the authors’ response: I am not convinced this is necessarily the process occurring in the animals’ brain. An alternative interpretation could be that the animals form a direct

association between a given stimulus and the action (or sequence of actions) it affords—without explicitly “counting” or transforming a visual numerical representation into a motor plan. The concept of “transformation” evokes well-known sensorimotor processes described for object grasping, where a visual description of 3D properties is converted into motor synergies configuring the hand and fingers appropriately (see Schaffelhofer and Scherberger, 2015, eLife). Translated into the numerical domain, this would imply a counting process, which indeed the authors aim to address (see line 81, “sensorimotor counting transformation task”). But is there clear evidence that these are truly “numerical cues” to be transformed, rather than more general abstract cues linked by associative learning to specific motor sequences?

Specific points:

1. Behavioral effects of format – Accuracy is maximal for $n = 1$ (no difference between signs and dots) and decreases from $n = 2$ to 5, with consistently higher accuracy for signs over dots. This effect should be statistically tested and discussed, as it might indicate that a single symbolic sign recruits the action representation more rapidly than a configuration of multiple dots ($n > 1$). While number-selective neurons show (by definition) no main or interaction effect involving “format,” a table reporting ANOVA classifications for all recorded neurons (including format-selective and task-unrelated types) would be useful. It would also be interesting to compare the temporal profile of neuronal discharge (and selectivity) by preferred numerosity—figure 3B shows a huge variability and single neuron examples suggests that neurons tuned to lower numerosities might respond earlier than those tuned to higher numerosities. Is this a genuine trend or just due to individual examples?
2. Temporal effects of stimulus format – Even without a main effect of “format” for number-selective neurons, a comparison of discharge (and possibly selectivity) onset for dots vs. signs would be informative. Given the behavioral advantage for signs, does neuronal activity also emerge earlier for signs?
3. Decoding by neuron type – To better quantify the role of number-selective neurons, compare decoding accuracy for the full VIP population (Fig. 5A) with that obtained using only number-selective or only number-unselective neurons. I would expect little or no difference for $n = 1$ (solvable via simple visuomotor association), but results for higher numbers might be very interesting.
4. Dynamics in figure 6B – The representation of number increases in accuracy as execution approaches, but the pattern is not strictly diagonal: the accuracy increases the closer it is to the execution phase. How would this look if $n = 1, 2$, etc. were plotted separately? Does the apparent dynamics reflect a general encoding principle or a bias from small (e.g. $n=1$) trials?
5. Evolutionary relevance – A brief discussion of the possible evolutionary significance of such a mechanism would be valuable, especially to clarify whether it reflects a natural property of the primate brain or a byproduct of extensive reinforcement learning.

Methods:

The task description is clear, but given its complexity, details on training would be important: How many steps were involved in the animal training procedure? In what order (e.g., fixation first, then dots, then Arabic numerals)? What reinforcement schedule was used at each stage?

Minor:

Figure 6B would be easier to read if the Y-axis values were inverted.

Reviewer #3

(Remarks to the Author)

Overall, Seidler and colleagues present a clear and concise manuscript demonstrating neurons in macaque ventral intraparietal cortex (VIP) tuned to the number of movements the animal has planned, before this plan is executed. Neurons here and nearby have previously been shown to respond to the number order of the current movement in an ongoing sequence (Sawamura et al., 2002) and the number of visual objects (numerosity) (the supervisor’s own lab). This is the first study I know of that demonstrates a response to the number of planned movements.

Methodologically, the study is generally sound with thorough controls that clearly demonstrate the neural response follows the number of planned movements, rather than the timing of the movements which would normally be strongly linked to the number. It is also clear that the neural response follows the number of planned movements rather any property of the cue. The neural response is also shown to be clearly relevant to behaviour, as error trials show responses to the number of movements the animal actually makes. These are all very strong points and follow the lab’s usual thorough methods.

There are, however, a few points that should be improved before publication. First, the response is often described in terms of a sensorimotor transformation (including in the first word of the title) and I don’t believe this is demonstrated. Instead, a response to a motor plan is clearly demonstrated. Second, there are a couple of circularities in some of the analyses. However, it seems likely these can be easily addressed and features of the data suggest the main conclusions will still be supported after they are addressed.

Major issues:

1) The study repeatedly and prominently describes the neural responses in terms of sensorimotor transformations. However, to me they only show a neural response to the motor plan. This motor plan is thoroughly trained in the animals, so it seems completely sufficient for the cues to trigger a motor plan without the animal doing any meaningful sensorimotor

transformation (for example they are not synchronising their movements to the cues). The neural responses also clearly generalise across cue formats (symbolic and non-symbolic numbers), and it is convincingly shown that the format of the cue (the sensory input) does not affect the neural response (driving the motor output). It seems possible that the animal perceives the cued number using their visual system, decides on a motor plan elsewhere (like a frontal executive system) and then plans how to do implement that elsewhere (like the premotor cortex). This study simply shows that a feature of the resulting motor plan (number) is then represented in VIP. I do not agree that the continuous and changing classification (Fig 6c) demonstrates a “continuous transition from sensory numerosity representation into a motor planning code for the upcoming number of hand movements”: this just shows a variety of response latencies and a dynamic neural code. While we know VIP contains neurons responding to visual numerosity (one possible sensory input), these are not examined here and (in humans) we do not see the same visually-driven responses to symbolic number like we see here. Without clearly linking the neural responses to the sensory input, the observed neural responses should simply be described in terms of a motor plan. The sensorimotor transformation aspect can still be considered in the discussion.

2) Demonstrating number-tuned responses is central to the study’s conclusions. However, this is done in a circular way. The number-tuned neurons are identified as those responding differently to different numbers, and then the preferred number is identified as that producing the largest response. Then, the average response is taken among the neurons with the same preferred number and a tuning curve is plotted (Figure 4a). The peak of the tuning curve is at a specific number, which is of course the number used to select the neurons. These neurons peak at that number because they were selected to peak at that number, i.e. circularity. I know this same analysis has been used extensively in Prof Nieder’s lab, but it is easily avoided and this should be done to make clear that the neurons are tuned when no circularity is used. The subsequent decoding analyses make clear that each neuron has multiple trials with the same number. A training set of these trials should be used to group neurons by their preferred number. An independent test set should then be used to quantify the response functions and plot the tuning functions for Fig 4a. The descriptive analysis in Fig 4b need not change. The ‘correct’ data in Fig 4c and Fig 4d should be taken from the test set, but the ‘error’ set already comes from independent trials: if both came from independent trials the comparison would be clearer.

3) The analyses presented in Figure 3 have a similar problem of circularity. First, the responsive time of the neuron is taken from the same data used to identify the response during that time. Of course, the responses to different numerosities differ within the identified time window, because that is how the time window was identified using the same data. And of course, Fig 3b shows the time windows progressing through the population of neurons, because the neurons are ordered by that time window in the same data. If the response time window is a real and repeatable property of the neurons (Figure 6 suggests it is) then the time window should be identified from one split of the data (like a “training” set) and then responses in an independent split of the data (test data) should be used to make Fig 3b. Likewise for Figures 3c-g, the time window and preferred numbers should come from the “training” set and the response amplitudes should come from the test set.

To be clear, in both the cases described in my 2nd and 3rd issues, I do not believe the results arise from circular analyses alone. The change in response amplitudes for numbers other than the preferred number suggest tuning. The decoding analyses suggest a repeatable time window and a repeatable response to number. Simply, circular analyses seem unnecessary and are problematic: the study would be better without them.

Minor issues: (Much of this is related to my 1st major issue)

-The description of the error trials in the results states that: “Together, these findings indicate that the sensorimotor population code in VIP reliably represents perceived numerosity and encodes both planned actions and the outcomes of number production, including over- or underestimation errors.”

I don’t agree that this is a representation of perceived numerosity. It is still the planned number of actions, and the error in that plan. We don’t know what the animal perceived, we only know its behaviour. Its behaviour follows a motor plan.

-In the description of generalisation across cue formats, the results sections states: “Because this modulation was independent of the instruction’s sensory format (dot versus sign), it indicates the presence of abstract, number-selective neurons involved in sensorimotor transformation.”

I don’t agree with this. We are seeing a response to the motor plan.

Signed,
Ben Harvey

Reviewer #4

(Remarks to the Author)

In this study, Seidler and colleagues investigate the role of the monkey parietal area VIP in linking numerical perception with action planning. Building on the well-established notion that VIP is involved in sensorimotor transformations, they employed a clever behavioral paradigm in which numerical cues (presented in different formats) instructed monkeys to plan and execute a corresponding number (from 1 to 5) of identical movements (releasing a handle). Two control paradigms were used to rule out potential confounding factors that might influence neural activity: movement time expectancy and total task duration/reward expectation.

The authors analyzed neural discharge during the preparation period at both the single-neuron and population levels, also applying decoding analyses. They conclude that VIP neurons are involved in converting abstract numerical input into goal-directed motor output (what they term the “sensorimotor foundation of numerical cognition”).

This is a solid study conducted by a leading group in the field of numerical representation in the brain. I found the topic addressed both original and broadly relevant. The task is well-designed, the methods are described in sufficient detail, the

analyses are sound, and the results are clear. The main weakness lies in the decision to focus the analysis exclusively on the planning phase (see major point 1). The discussion is well centered on the data, although it could benefit from being placed within a richer theoretical framework -particularly by considering other phases of the task. Below I list my concerns and suggestions to further improve the manuscript.

Major points:

1) I place this point first because I believe it is the most interesting, although I do not require a response to it as mandatory. The authors appear to have missed an opportunity to explore the broader neural substrate of numerical coding. The study focuses exclusively on a single task phase -the motor planning period- without examining what happens earlier (e.g., upon presentation of the instructing stimulus) or later (e.g., during motor execution and performance feedback). Interestingly, individual neurons show strong activation after the planning phase (Fig. 3C, D, F; note that activity in E and G is truncated). This aligns with previous literature and becomes even more apparent when considering population activity (Fig. 6A-C). In particular, Fig. 6B suggests a static pattern that might indicate a shared coding scheme between the late cueing phase and action execution (see circles in my figure).

It would be very interesting to assess, at both the single-neuron and population levels, the activity during the execution of each movement -especially the final one. Such analyses could reinforce the already compelling findings regarding the planning phase while also offering new insights into how numbers might be encoded in the parietal cortex. A possible hypothesis is that, similar to what has recently been proposed for instructions and contextual cues in the prefrontal cortex, the parietal lobe could engage in "pragmatic coding" -in which numbers are not just triggers for motor sequences but are themselves encoded (within VIP) as intended movements. This is especially relevant given the authors' claim at lines 167-168: "This temporal profile indicates an abstract, population-level representation of numerosity in VIP neurons during the sensorimotor transformation from perception to action." Including analyses from the cue and motor execution phases would also be valuable in the context of error trials, especially considering the strong alignment between neuronal and behavioral data.

2) Where were the neurons recorded from, exactly? Fig. 3A shows a coronal section highlighting VIP, but is this based on histological reconstruction or on a priori MRI scans? Were the electrodes inserted through guide tubes placed at depth, or lowered freely from the cortical surface to a depth of 8–12 mm? Were the entry points located on the dorsal or ventral bank? Additional methodological details are needed to confidently attribute the recordings to VIP. Otherwise, the results should be described more generally as pertaining to the anterior intraparietal region (e.g., VIP, PEip, AIP?).

3) The negative results from the PEV analysis regarding stimulus format are very interesting ("The neuronal population encoded numerosity significantly, while neither stimulus format nor the interaction between the two showed significant encoding"), but they should be further validated at the single-neuron level. The first single-neuron analysis uses a two-factor sliding window ANOVA, but only the results for the numerosity factor are described. The authors should briefly report the findings for the format factor as well.

4) Were neurons characterized based on their sensory and/or motor properties?

5) Methods: "To construct population response functions, firing rates were normalized within each neuron to its minimum and maximum response and then averaged across neurons grouped by their preferred numerosity." How many inhibited neurons were included in this normalization? For instance, neuron 3F appears inhibited during this phase, was its most inhibited state considered as its "preferred numerosity" in this case?

Minor points

- Given the emphasis on population analyses, the authors could add a figure showing mean time course activity. This would be valuable, as in many cases the neuronal activity -though clearly modulated during the analyzed epoch- peaks in other phases.

- Letters in the text and figures are inconsistently labeled (e.g., "B" in the figure vs. "b" in the text).

- Fig. 3A: neuronal activity is truncated differently in panels B, C, D, F compared to E and G.

Version 1:

Reviewer comments:

Reviewer #1

(Remarks to the Author)

Now the authors seem to have responded properly to all my comments. I have no more to add.

Reviewer #2

(Remarks to the Author)

I thank the authors for the new analyses and for their thorough responses to my original comments, which help provide stronger support for the hypothesis that a sensorimotor transformation (rather than a simple association) may indeed play a specific role in task execution.

Reviewer #3

(Remarks to the Author)

My concerns are fully addressed in the revised version.

Reviewer #4

(Remarks to the Author)

The authors responded to all my critics and clarified my doubts.

I believe the work has been improved and deserves the publication in the present form.

REPLY TO REVIEWER COMMENTS (1. Revision)

Reviewer #1 (Remarks to the Author):

In this article by Seidler et al., the authors investigated the neural mechanisms for transformation of perceived numerical values into corresponding numbers of self-generated actions. For this purpose, they trained the monkeys to perform the task in which the subjects viewed visual numerical cues and produced a corresponding number of hand movements. They recorded the single neuron activities from VIP and found neurons tuned to the number of intended actions. Population decoding confirmed that these VIP neurons encoded intended number of actions including over- and underestimated errors. Thus, the neuronal mechanisms for conversion of abstract numerical input to actions were clarified for the VIP neurons. The study was well designed, and results are clear. But I have to make some comments about some unclear documentation to make this manuscript more concise to the readers.

We thank the reviewer for recognizing the clarity and design of our study. We also appreciate the constructive suggestion regarding the manuscript's documentation. We have revised the text to clarify these points and make the presentation more concise and accessible to readers.

Major comments:

1. Fig. 1C shows that a variety of intervals between each wait period were tested. The authors may wish to show that such variation did not affect the neural code of numerical values. But how was the result? At least, the authors should show some quantitative evidence that such manipulation did not affect the results.

Thank you for pointing this out. We apologize for any confusion in our description of the neuronal data. The temporal variations shown in Fig. 1C occurred *after* the motor preparation period, during the period when monkeys were required to execute the corresponding number of hand movements. To ensure that monkeys were not using timing cues, the total duration and rhythmicity of hand movements were controlled in the control conditions. Importantly, all neuronal analyses reported in the manuscript are based on the *motor planning period preceding any hand movements*, and the monkeys were unaware of whether a trial belonged to the standard or control conditions. Therefore, the temporal arrangement of the motor execution intervals could not have influenced the neuronal coding of numerical values during the motor planning period.

We have clarified this point in the revised manuscript in the figure legend 1C (**line 561ff**), the results (**line 79ff**), and the methods (**lines 341ff & 346ff**).

2. As I see the example data in Fig. 3, the temporal profile of the neuronal responses in panels C-G are quite variant. There, the peak of the activity seems to be earlier for the neurons encoding smaller numbers. Is it true, or what the authors meant? Then, how would the panel B look like, if individual neurons in the panel are indicated with the colors corresponding to the best preferred number?

The reviewer is correct that the temporal profiles of the neurons are variable, which is common for selective neurons in VIP. The example neurons shown were chosen arbitrarily and are not meant to suggest any systematic latency differences related to the encoded number.

Following the reviewer's suggestion, we have **updated Fig. 3B** to include a surface plot with color bars indicating the selective time intervals for each neuron according to its preferred number. As shown, the results are randomly distributed, with no systematic temporal ordering of activity based on preferred number. We added this information in the results (**line 123f**).

3. Static vs dynamic codes. The data in Fig. 6 shows that there are both static and dynamic codes. If these are dissociable, the readers may wonder how the static and dynamic codes appear for each neuron class with different preference for the number. I wonder whether the authors can show this kind of data.

We thank the reviewer for this insightful suggestion. We do not see a principled way to separate neurons with static versus dynamic codes, as these represent extremes along a continuum rather than distinct categories.

However, to address this point, we performed an additional cross-temporal decoding analysis by grouping neurons according to their preferred numerosity into two groups: numerosity-1 preferring neurons and numerosity-2–5 preferring neurons. Given the relatively small fraction of neurons tuned to numerosities 2–5 (see Fig. 4B), it was not possible to perform the decoding analysis for each numerosity class individually.

We found that the cross-temporal decoding matrices were very similar across the two groups, indicating a mixture of both static and dynamic codes for all number classes (**new Supplementary Fig. S4**). These results suggest that static and dynamic coding mechanisms contribute across the entire population of number-selective neurons, regardless of their specific preferred numerosity. This point has now been added to the Results section (**lines 232f**).

Minor comments:

4. I feel some discomfort in calling different kind of format in Fig. 1B “standard” and “control”. They were prepared just for comparison to exclude the possible involvement of non-numerical factors. I feel more confirmable to label them “format 1” and “format 2” etc.

We appreciate the reviewer’s concern regarding the terminology used for the different stimulus conditions. We agree that labeling experimental conditions is rarely entirely unambiguous. In the present study, however, we prefer to retain the labels “standard” and “control” that are defined explicitly in the Methods for the following reasons.

First, the “standard” condition randomizes all non-numerical visual parameters, including item size, spatial extent, density, and inter-item distance. It therefore produces the most variable displays obtained by systematically shuffling these parameters and serves as a baseline reference condition. The “control” condition, in contrast, explicitly controls for total dot area and mean inter-item distance across numerosities, thus functioning as a true control for these non-numerical factors.

Second, we have consistently used these labels for nearly 25 years across studies in monkeys, crows, and humans. Maintaining the same terminology facilitates direct comparison across experiments, species, and laboratories. For reasons of continuity and comparability with the existing literature, we therefore believe that retaining these labels is beneficial.

5. Rightmost part of the data in Fig, 3E and G might have been missing. Please show the full dataset.

We apologize for this truncation and thank the reviewer for noticing it. The omission was due to a figure-formatting error. This issue has now been corrected, and the complete datasets are shown in the revised Fig. 3E and 3G.

Reviewer #2 (Remarks to the Author):

The goal of this paper is clear and straightforward: to test whether neurons in the VIP encode the transformation of visually perceived numerosity into the corresponding number of planned motor actions. The introduction situates the research question well within the existing literature, and the issue addressed is clearly novel. The manuscript is well written and pleasant to read.

However, I have a single major concern regarding the interpretation of the findings. In my view, the key sentence appears on page 4, line 61: “the monkeys translated the perceived numerical value into a planned sequence of corresponding hand movements”. I am not an expert in this specific field, but I would like the authors’ response: I am not convinced this is necessarily the process occurring in the animals’ brain. An alternative interpretation could be that the animals form a direct association between a given stimulus and the action (or sequence of actions) it affords—without explicitly “counting” or transforming a visual numerical representation into a motor plan. The concept of “transformation” evokes well-known sensorimotor processes described for object grasping, where a visual description of 3D properties is converted into motor synergies configuring the hand and fingers appropriately (see Schaffelhofer and Scherberger, 2015, eLife). Translated into the numerical domain, this would imply a counting process, which indeed the authors aim to address (see line 81, “sensorimotor counting transformation task”). But is there clear evidence that these are truly “numerical cues” to be transformed, rather than more general abstract cues linked by associative learning to specific motor sequences?

We thank the reviewer for this thoughtful and important comment. We agree that, in principle, the observed behavior could be explained by a learned association between abstract cues and specific motor sequences, without requiring an explicit transformation of a sensory numerical representation into a motor plan. For this reason, we have substantially extended our analyses to directly test whether numerical information present during sensory processing is reflected, maintained, and reformatted during motor preparation within VIP, rather than merely triggering a precomputed motor plan elsewhere.

Specifically, we now show that numerical selectivity is already present during the sensory instruction phase and not confined to motor preparation. Approximately 10% of VIP neurons are number selective during instruction cue presentation, consistent with previous VIP recordings, and this selectivity persists into the motor planning phase (**new Table 2**). Although the proportion of selective neurons increases during planning, the presence of number tuning during the sensory phase argues against a purely motor-only representation.

In addition, time-resolved population decoding reveals that numerical information can be continuously decoded from VIP activity beginning in the second half of the instruction phase and extending smoothly into the motor preparation phase, without an interruption between phases (**new Fig. 6A,B**). Across-phase decoding further demonstrates that classifiers trained on sensory-phase activity generalize significantly above chance to motor preparation activity, and vice versa (**new Fig. 6C**). Finally, time-resolved SVM decoding (**Fig. 8C**) shows number information rising systematically from the second half of the instruction period and persisting into motor preparation. All this indicates that numerical information is shared across phases rather than being recomputed de novo as a motor plan.

Together, these findings provide evidence for continuity between sensory and motor representations of number in VIP and support the interpretation that numerical information conveyed by the instruction stimulus is transformed into a preparatory motor code within this area. While we do not claim to demonstrate explicit “counting” in the strict algorithmic sense, the data go beyond a simple stimulus–response association and instead point to an intermediate sensorimotor representation of numerosity.

These new analyses are presented in section “**Temporal evolution of sensory-to-motor population decoding of number information**” in the Results (lines 162ff)

In addition, we have revised the manuscript accordingly and now discuss this issue in more detail (**lines 292ff**), including citations of Schaffelhofer and Scherberger (2015, eLife) and several other pertinent monkey studies, in the context of related work on sensorimotor mapping in the intraparietal sulcus.

Specific points:

1. Behavioral effects of format – Accuracy is maximal for $n = 1$ (no difference between signs and dots) and decreases from $n = 2$ to 5, with consistently higher accuracy for signs over dots. This effect should be statistically tested and discussed, as it might indicate that a single symbolic sign recruits the action representation more rapidly than a configuration of multiple dots ($n > 1$).

While number-selective neurons show (by definition) no main or interaction effect involving “format,” a table reporting ANOVA classifications for all recorded neurons (including format-selective and task-unrelated types) would be useful. It would also be interesting to compare the temporal profile of neuronal discharge (and selectivity) by preferred numerosity—figure 3B shows a huge variability and single neuron examples suggests that neurons tuned to lower numerosities might respond earlier than those tuned to higher numerosities. Is this a genuine trend or just due to individual examples?

We added the precise percentage-correct data in a **new Table 1** and included the corresponding statistical analyses in the Results section (**line 90ff**), together with an expanded discussion (**line 282ff**). Both monkeys performed better with sign formats than with dot numerosities. We suspect that this difference reflects the higher visual variability and perceptual noise inherent in dot numerosity displays, whereas the sign stimuli serve as clear instruction cues that signal numerical information more reliable.

Concerning selective neurons, we now included **Table 2** that lists the numbers of neurons selective to main factors and interactions. Following the reviewer’s suggestion, we have **updated Fig. 3B** to include a surface plot with color bars indicating the selective time intervals for each neuron according to its preferred number. As shown, the results are randomly distributed, with no systematic temporal ordering of activity based on preferred number. We added this information in the results (**line 124ff**).

2. Temporal effects of stimulus format – Even without a main effect of “format” for number-selective neurons, a comparison of discharge (and possibly selectivity) onset for dots vs. signs would be informative. Given the behavioral advantage for signs, does neuronal activity also emerge earlier for signs?

We compared the onset of numerical selectivity for dot and sign stimuli and found no significant difference between formats (Results, **lines 127ff**). Thus, despite the behavioral advantage for signs, neuronal selectivity in VIP emerged at comparable latencies for both stimulus types.

3. Decoding by neuron type – To better quantify the role of number-selective neurons, compare decoding accuracy for the full VIP population (Fig. 5A) with that obtained using only number-selective or only number-unselective neurons. I would expect little or no difference for $n = 1$ (solvable via simple visuomotor association), but results for higher numbers might be very interesting.

We agree that this is an excellent suggestion. We now directly compare decoding performance obtained from the full VIP population with that based exclusively on numerosity-selective neurons and, separately, on numerosity-unselective neurons during the motor planning period. As shown in the **new Supplementary Fig. S3A**, decoding accuracy is substantially higher when using numerosity-selective neurons than when using numerosity-unselective neurons, (**new Supplementary Fig. S3B**) demonstrating that number-selective cells make a critical

contribution to population-level decoding of target numerosity. These new results are described in the Results section (**lines 154ff**).

4. Dynamics in figure 6B – The representation of number increases in accuracy as execution approaches, but the pattern is not strictly diagonal: the accuracy increases the closer it is to the execution phase. How would this look if $n = 1, 2$, etc. were plotted separately? Does the apparent dynamics reflect a general encoding principle or a bias from small (e.g. $n=1$) trials?

We agree that the non-diagonal structure in now **Fig. 8B** raises the question of whether the observed dynamics reflect a general encoding principle or are biased by small numerosities ($n = 1$). To address this, we performed an additional cross-temporal decoding analysis in which neurons were grouped according to their preferred numerosity. One group consisted of numerosity-1–preferring neurons, and the second group comprised neurons preferring numerosities 2–5. Because the proportion of neurons tuned to individual numerosities 2–5 was relatively small (**Fig. 4B**), separate decoding analyses for each numerosity were not feasible.

The resulting cross-temporal decoding matrices were highly similar for both groups (**new Supplementary Fig. S4**), showing comparable mixtures of static and dynamic coding. This indicates that the increase in decoding accuracy toward the execution phase is not driven specifically by numerosity-1 trials, but reflects a general population-level encoding principle that applies across different numerical categories. We now report this result in the Results section (**lines 273f**).

5. Evolutionary relevance – A brief discussion of the possible evolutionary significance of such a mechanism would be valuable, especially to clarify whether it reflects a natural property of the primate brain or a byproduct of extensive reinforcement learning.

We thank the reviewer for this suggestion. We have extended the discussion to address the possible evolutionary significance of a sensorimotor number transformation mechanism and clarified that it likely reflects a natural property of the primate brain rather than a byproduct of extensive reinforcement learning. A new paragraph has been added in the Discussion section (**lines 347ff**).

Methods:

The task description is clear, but given its complexity, details on training would be important: How many steps were involved in the animal training procedure? In what order (e.g., fixation first, then dots, then Arabic numerals)? What reinforcement schedule was used at each stage?

We thank the reviewer for this suggestion. We have added a new paragraph describing the animal training procedure in detail, including the sequential training steps, stimulus order (fixation, dot stimuli, then Arabic numerals), and the reinforcement schedule at each stage. This information is now included in the Methods section (**lines 416ff**).

Minor:

Figure 6B would be easier to read if the Y-axis values were inverted.

Thank you for the suggestion. We have inverted the Y-axis in Figure 6B to improve readability.

Reviewer #3 (Remarks to the Author):

Overall, Seidler and colleagues present a clear and concise manuscript demonstrating neurons in macaque ventral intraparietal cortex (VIP) tuned to the number of movements the animal has planned, before this plan is executed. Neurons here and nearby have previously been shown to respond to the number order of the current movement in an ongoing sequence (Sawamura et al., 2002) and the number of visual objects (numerosity) (the supervisor's own lab). This is the first study I know of that demonstrates a response to the number of planned movements.

Methodologically, the study is generally sound with thorough controls that clearly demonstrate the neural response follows the number of planned movements, rather than the timing of the movements which would normally be strongly linked to the number. It is also clear that the neural response follows the number of planned movements rather any property of the cue. The neural response is also shown to be clearly relevant to behaviour, as error trials show responses to the number of movements the animal actually makes. These are all very strong points and follow the lab's usual thorough methods.

There are, however, a few points that should be improved before publication. First, the response is often described in terms of a sensorimotor transformation (including in the first word of the title) and I don't believe this is demonstrated. Instead, a response to a motor plan is clearly demonstrated. Second, there are a couple of circularities in some of the analyses. However, it seems likely these can be easily addressed and features of the data suggest the main conclusions will still be supported after they are addressed.

We thank Dr. Harvey for the careful and thoughtful review of our manuscript and for highlighting the strengths of our study. We appreciate the constructive suggestions and have revised the manuscript accordingly.

Major issues:

1) The study repeatedly and prominently describes the neural responses in terms of sensorimotor transformations. However, to me they only show a neural response to the motor plan. This motor plan is thoroughly trained in the animals, so it seems completely sufficient for the cues to trigger a motor plan without the animal doing any meaningful sensorimotor transformation (for example they are not synchronising their movements to the cues). The neural responses also clearly generalise across cue formats (symbolic and non-symbolic numbers), and it is convincingly shown that the format of the cue (the sensory input) does not affect the neural response (driving the motor output). It seems possible that the animal perceives the cued number using their visual system, decides on a motor plan elsewhere (like a frontal executive system) and then plans how to do implement that elsewhere (like the premotor cortex). This study simply shows that a feature of the resulting motor plan (number) is then represented in VIP. I do not agree that the continuous and changing classification (Fig 6c) demonstrates a "continuous transition from sensory numerosity representation into a motor planning code for the upcoming number of hand movements": this just shows a variety of response latencies and a dynamic neural code. While we know VIP contains neurons responding to visual numerosity (one possible sensory input), these are not examined here and (in humans) we do not see the same visually-driven responses to symbolic number like we see here. Without clearly linking the neural responses to the sensory input, the observed neural responses should simply be described in terms of a motor plan. The sensorimotor transformation aspect can still be considered in the discussion.

We thank the reviewer for this insightful comment. We agree that it is important to explicitly link neural responses during the motor planning phase to the sensory input (or vice versa) in order to demonstrate that a transformation of sensory numerosity representations into a motor planning code occurs within VIP. Such evidence is necessary to show that VIP neurons do

more than simply reflect a motor plan inherited from other areas. We are now adding new analyses to support our claim of a transformation:

A) We now compare number tuning during the sensory epoch (instruction cue presentation) and the motor planning epoch. We report the proportion of number-selective neurons in each epoch in the **new Table 2 (line 111)**. We find a significant proportion of sample-tuned neurons (approximately 10%), which is consistent with our previous recordings from VIP. Although the proportion of number-selective neurons is higher during the motor planning period, selectivity is present not only during planning but also during the sensory epoch. This pattern argues that VIP neurons do more than merely represent a motor plan; rather, they have the capacity to transform numerical sensory input into a motor planning code.

B) We performed a new classifier decoding analyses using sliding time windows spanning fixation, instruction, and motor preparation phases to evaluate the continuity of number coding. The resulting confusion matrices are shown in the **new Figure 6A**. They demonstrate that a classifier trained on population activity of VIP neurons can continuously decode the instructed number, beginning in the second half of the instruction stimulus phase and continuing with increasing accuracy throughout the motor planning period, without an interruption between the instruction and planning phases. This smooth transition from instruction-related to motor-related decoding is further supported by the series of classifier tuning curves for each analyzed time window (**new Fig. 6B**), which show that decoding of the instructed—and impending—number increases continuously until the end of the planning phase, as expected for a sensory-to-motor transformation process.

C) We investigated across-phase decoding of numerical information by training a classifier on data from the sensory epoch and testing it on the motor planning epoch, and vice versa. The results are presented in the **new Figure 6C**. The classifier trained on the sensory (instruction) phase successfully generalizes the motor planning phase. As expected, decoding performance decreases when transferring across epochs; importantly, however, it remains significantly above chance. These findings support the notion that numerical information carried by VIP neurons is transferred from the sensory representation into the motor planning phase. In other words, they demonstrate continuity between sensory and motor representations and argue against a purely motor-based readout.

D) The time-resolved SVM decoding (**Fig. 8C**) similarly reveals number information increasing from the latter half of the instruction period into motor preparation.

Together, these findings based on new analyses suggest that sensory numerosity information conveyed by the instruction stimulus is transformed inside VIP into a preparatory motor signal representing the impending number of hand movements.

These new analyses are presented in section “**Temporal evolution of sensory-to-motor population decoding of number information**” in the Results (**lines 169ff**). In addition, we are now discussing this question in more detail (**lines 296ff**), also in the light of other studies reporting sensorimotor mapping in the intraparietal sulcus.

2) Demonstrating number-tuned responses is central to the study’s conclusions. However, this is done in a circular way. The number-tuned neurons are identified as those responding differently to different numbers, and then the preferred number is identified as that producing the largest response. Then, the average response is taken among the neurons with the same preferred number and a tuning curve is plotted (Figure 4a). The peak of the tuning curve is at a specific number, which is of course the number used to select the neurons. These neurons peak at that number because they were selected to peak at that number, i.e. circularity. I know this same analysis has been used extensively in Prof Nieder’s lab, but it is easily avoided and this should be done to make clear that the neurons are tuned when no circularity is used. The subsequent decoding analyses make clear that each neuron has multiple trials with the same number. A training set of these trials should be used to group neurons by their preferred

number. An independent test set should then be used to quantify the response functions and plot the tuning functions for Fig 4a. The descriptive analysis in Fig 4b need not change. The ‘correct’ data in Fig 4c and Fig 4d should be taken from the test set, but the ‘error’ set already comes from independent trials: if both came from independent trials the comparison would be clearer.

We thank the reviewer for this suggestion and performed the recommended split-half cross-validation of tuning curves. Trials for each neuron were divided into balanced training and test sets (50/50) using an even-versus-odd split. The preferred number was determined from one half of the data, and tuning curves were computed from the other half.

The details and results of this analysis are shown in the **new Supplementary Figure S2** and described in the Methods (**lines 525ff**). The results are highly stable and are now reported in the Results (**lines 139ff**). For example, the median correlation coefficient between the preferred numbers in the training and test sets was 0.91. Despite the reduction in data due to the split, the tuning functions remained exceedingly stable, providing strong support for our conclusions.

3) The analyses presented in Figure 3 have a similar problem of circularity. First, the responsive time of the neuron is taken from the same data used to identify the response during that time. Of course, the responses to different numerosities differ within the identified time window, because that is how the time window was identified using the same data. And of course, Fig 3b shows the time windows progressing through the population of neurons, because the neurons are ordered by that time window in the same data. If the response time window is a real and repeatable property of the neurons (Figure 6 suggests it is) then the time window should be identified from one split of the data (like a “training” set) and then responses in an independent split of the data (test data) should be used to make Fig 3b. Likewise for Figures 3c-g, the time window and preferred numbers should come from the “training” set and the response amplitudes should come from the test set.

To be clear, in both the cases described in my 2nd and 3rd issues, I do not believe the results arise from circular analyses alone. The change in response amplitudes for numbers other than the preferred number suggest tuning. The decoding analyses suggest a repeatable time window and a repeatable response to number. Simply, circular analyses seem unnecessary and are problematic: the study would be better without them.

To address the concern, we split the data by stimulus format (dots vs. signs). The response time windows were determined from one format (e.g., dots) and can be compared the other format (e.g., signs). The data are presented in **Supplementary Fig. S1**. This approach confirms that the response windows are reliable and not an artifact of selecting from the same trials. We note that fully halving the data to create independent training and test sets would introduce unnecessary noise due to the smaller trial counts.

Minor issues: (Much of this is related to my 1st major issue)

-The description of the error trials in the results states that: “Together, these findings indicate that the sensorimotor population code in VIP reliably represents perceived numerosity and encodes both planned actions and the outcomes of number production, including over- or underestimation errors.”

I don’t agree that this is a representation of perceived numerosity. It is still the planned number of actions, and the error in that plan. We don’t know what the animal perceived, we only know its behaviour. Its behaviour follows a motor plan.

We are surprised that this point is considered controversial. Analyses of the sensory instruction phase reveal VIP neurons tuned to dot number and to symbolic numerical values, many of which maintain this tuning into the motor preparation phase. Perhaps these new analyses in the revision are more convincing. In this task, numerosity must be extracted from the sensory input before a motor plan can be formed; otherwise the animal would have no basis for selecting the number of movements. Thus, behavior necessarily reflects perceived numerosity. Together, these findings support the interpretation that VIP activity reflects numerosity as it is transformed into action, rather than motor output alone.

-In the description of generalisation across cue formats, the results sections states: “Because this modulation was independent of the instruction’s sensory format (dot versus sign), it indicates the presence of abstract, number-selective neurons involved in sensorimotor transformation.”

I don’t agree with this. We are seeing a response to the motor plan.

With the same reasoning as above, we respectfully disagree. We identified VIP neurons tuned to dot number and symbolic numerical values during the sensory instruction phase, many of which maintain this tuning into motor preparation. Because numerosity must be extracted from the sensory input before a motor plan can be formed, the cue-independent modulation reflects numerosity transformed into action, not motor planning alone. The new analyses of the sensory instruction phase (**Fig. 6**) more directly address the sensory-to-motor transition and therefore strengthen this interpretation.

Reviewer #4 (Remarks to the Author):

In this study, Seidler and colleagues investigate the role of the monkey parietal area VIP in linking numerical perception with action planning. Building on the well-established notion that VIP is involved in sensorimotor transformations, they employed a clever behavioral paradigm in which numerical cues (presented in different formats) instructed monkeys to plan and execute a corresponding number (from 1 to 5) of identical movements (releasing a handle). Two control paradigms were used to rule out potential confounding factors that might influence neural activity: movement time expectancy and total task duration/reward expectation.

The authors analyzed neural discharge during the preparation period at both the single-neuron and population levels, also applying decoding analyses. They conclude that VIP neurons are involved in converting abstract numerical input into goal-directed motor output (what they term the “sensorimotor foundation of numerical cognition”).

This is a solid study conducted by a leading group in the field of numerical representation in the brain. I found the topic addressed both original and broadly relevant. The task is well-designed, the methods are described in sufficient detail, the analyses are sound, and the results are clear. The main weakness lies in the decision to focus the analysis exclusively on the planning phase (see major point 1). The discussion is well centered on the data, although it could benefit from being placed within a richer theoretical framework -particularly by considering other phases of the task.

Below I list my concerns and suggestions to further improve the manuscript.

We thank the reviewer for their thoughtful evaluation and positive comments. We appreciate the suggestion to consider additional phases of the task and have now included analyses and discussion addressing neural activity during the sensory instruction and motor planning phases, which we believe strengthens the theoretical framing of the manuscript.

Major points:

1) I place this point first because I believe it is the most interesting, although I do not require a response to it as mandatory.

The authors appear to have missed an opportunity to explore the broader neural substrate of numerical coding. The study focuses exclusively on a single task phase - the motor planning period- without examining what happens earlier (e.g., upon presentation of the instructing stimulus) or later (e.g., during motor execution and performance feedback).

Interestingly, individual neurons show strong activation after the planning phase (Fig. 3C, D, F; note that activity in E and G is truncated). This aligns with previous literature and becomes even more apparent when considering population activity (Fig. 6A-C). In particular, Fig. 6B suggests a static pattern that might indicate a shared coding scheme between the late cueing phase and action execution (see circles in my figure).

It would be very interesting to assess, at both the single-neuron and population levels, the activity during the execution of each movement -especially the final one. Such analyses could reinforce the already compelling findings regarding the planning phase while also offering new insights into how numbers might be encoded in the parietal cortex. A possible hypothesis is that, similar to what has recently been proposed for instructions and contextual cues in the prefrontal cortex, the parietal lobe could engage in "pragmatic coding" -in which numbers are not just triggers for motor sequences but are themselves encoded (within VIP) as intended movements. This is especially relevant given the authors' claim at lines 167-168: “This temporal profile indicates an abstract,

population-level representation of numerosity in VIP neurons during the sensorimotor transformation from perception to action.”

Including analyses from the cue and motor execution phases would also be valuable in the context of error trials, especially considering the strong alignment between neuronal and behavioral data.

We are grateful for this comment and agree that linking the motor preparation period to the preceding instruction period provides a clearer picture, including with respect to the conceptual issues raised by the reviewers. Because we have previously published extensively on sample numerosity selectivity, this period was not included in the original version of the manuscript. However, we recognize the importance of incorporating it here and connecting this sensory phase to the motor planning phase.

In response to this comment, we now present numerosity selectivity at the level of single neurons (**Fig. 3B, new Table 2**) from instruction stimulus onset through the end of the motor planning period.

In addition, we analyzed classifier decoding accuracies using sliding time windows across fixation, instruction and motor preparation phases (**new Fig. 6A,B**). These analyses reveal a smooth transition of number decoding from the instruction phase into the motor planning phase.

We also performed across-phase decoding by training a classifier on the sensory epoch and testing on the motor planning epoch (and vice versa; **new Fig. 6C**). Decoding generalizes across phases and remains significantly above chance, indicating that numerical information in VIP neurons is carried from sensory input into motor planning. This demonstrates continuity between sensory and motor representations and argues against a purely motor-based readout.

In addition, the time-resolved SVM decoding (**Fig. 8C**) similarly reveals number information increasing from the latter half of the instruction period into motor preparation.

Together, these results suggest that sensory numerosity information conveyed by the instruction stimulus is transformed into a preparatory motor signal representing the impending number of hand movements, indicating that VIP neurons bridge sensory and motor planning phases.

We agree that analyzing neural activity during movement execution is an interesting and important direction. However, a detailed examination of the information represented during the execution period would require substantial and new analyses and would go beyond the scope of the present, already extensive manuscript. We therefore defer a comprehensive analysis of the execution phase to a dedicated follow-up study.

2) Where were the neurons recorded from, exactly? Fig. 3A shows a coronal section highlighting VIP, but is this based on histological reconstruction or on a priori MRI scans? Were the electrodes inserted through guide tubes placed at depth, or lowered freely from the cortical surface to a depth of 8–12 mm? Were the entry points located on the dorsal or ventral bank? Additional methodological details are needed to confidently attribute the recordings to VIP. Otherwise, the results should be described more generally as pertaining to the anterior intraparietal region (e.g., VIP, PEip, AIP?).

We appreciate this comment. The recording sites were determined based on individual *a priori* MRI scans from both monkeys, combined with reconstructed coordinates and electrode track depths. Electrodes were inserted transdurally from the cortical surface to depths of 8–12 mm, each attached to a custom-made mechanical microdrive. Entry points were located on both the dorsal and ventral banks to access VIP in the fundus of the intraparietal sulcus. In previous experiments, we found global visual motion direction selectivity and tactile responses at the target recording sites, consistent with the known functional properties of VIP. These

methodological details have been clarified in the revised Methods section (**lines 470ff**) to support the attribution of the recordings to VIP.

3) The negative results from the PEV analysis regarding stimulus format are very interesting (“The neuronal population encoded numerosity significantly, while neither stimulus format nor the interaction between the two showed significant encoding”), but they should be further validated at the single-neuron level.

The first single-neuron analysis uses a two-factor sliding window ANOVA, but only the results for the numerosity factor are described. The authors should briefly report the findings for the format factor as well.

Thank you for this suggestion. We now report the full ANOVA findings for the single neurons in **new Table 2**.

4) Were neurons characterized based on their sensory and/or motor properties?

In both the instruction stimulus phase and the motor preparation phase, neurons were characterized based on numerical values, i.e., abstract numerical categories. During the instruction stimulus phase, these categories were cued by dot displays and associated signs. During the motor preparation phase, they corresponded to the instructed number of hand movements. This is now clarified in **lines 508ff**.

5) Methods: “To construct population response functions, firing rates were normalized within each neuron to its minimum and maximum response and then averaged across neurons grouped by their preferred numerosity.”

How many inhibited neurons were included in this normalization? For instance, neuron 3F appears inhibited during this phase, was its most inhibited state considered as its “preferred numerosity” in this case?

We assessed how many numerosity-selective neurons showed excitatory versus inhibitory responses by comparing the average firing rate during the numerosity-selective interval (collapsed across all five numerosities) with baseline activity during the fixation period. Neurons were classified as excited if their activity exceeded baseline and as inhibited if it fell below baseline.

In the instruction stimulus phase, we found 12 numerosity-selective neurons were classified as excited and 9 as inhibited. In the motor preparation phase, 56 neurons were excited and 29 were inhibited. This information has now been added to **lines 534ff** in the Methods, and **lines 115ff** in the Results.

Minor points

- Given the emphasis on population analyses, the authors could add a figure showing mean time course activity. This would be valuable, as in many cases the neuronal activity -though clearly modulated during the analyzed epoch- peaks in other phases.

The mean time-course activity of numerosity-selective neurons is shown in **Fig. 8A** and **C**. It is quantified as percentage of explained variance (PEV) over the course of the trial in **Fig. 8A**, and as the mean decoding accuracy of a time-resolved SVM classifier in **Fig. 8C**.

- Letters in the text and figures are inconsistently labeled (e.g., “B” in the figure vs. “b” in the text).

Thank you for noticing this inconsistency. We have corrected all figure and text references to use capital letters throughout, in accordance with the journal's style guidelines.

- Fig. 3A: neuronal activity is truncated differently in panels B, C, D, F compared to E and G.

We apologize for this truncation and thank the reviewer for noticing it. The omission was due to a figure-formatting error. This issue has now been corrected, and the complete datasets are shown in the revised Fig. 3E and 3G.

REPLY TO REVIEWER COMMENTS (2. Revision)

REVIEWERS' COMMENTS

Reviewer #1 (Remarks to the Author):

Now the authors seem to have responded properly to all my comments.
I have no more to add.

Reviewer #2 (Remarks to the Author):

I thank the authors for the new analyses and for their thorough responses to my original comments, which help provide stronger support for the hypothesis that a sensorimotor transformation (rather than a simple association) may indeed play a specific role in task execution.

Reviewer #3 (Remarks to the Author):

My concerns are fully addressed in the revised version.

Reviewer #4 (Remarks to the Author):

The authors responded to all my critics and clarified my doubts.
I believe the work has been improved and deserves the publication in the present form.

We thank all reviewers for their thoughtful comments. We are pleased that our revisions and additional analyses have addressed your concerns and contributed to strengthening the manuscript.

C